# Spatial snapshots of amyloid precursor protein intramembrane processing via early endosome proteomics

Hankum Park [1,2,6], Frances V. Hundley [1,2,8], Qing Yu[1,8],
Katherine A. Overmyer [3,4], Dain R. Brademan[3,4], Lia Serrano[3], Joao A. Paulo [1],
Julia C. Paoli [1,2], Sharan Swarup[1,2,7], Joshua J. Coon [3,4,5], Steven P. Gygi [1] &
J. Wade Harper [1,2] ✉

Degradation and recycling of plasma membrane proteins occurs via the endolysosomal system, wherein endosomes bud into the cytosol from the plasma membrane and subsequently mature into degradative lysosomal compartments. While methods have been developed for rapid selective capture of lysosomes (Lyso-IP), analogous methods for isolation of early endosome intermediates are lacking. Here, we develop an approach for rapid isolation of early/sorting endosomes through affinity capture of the early endosome-associated protein EEA1 (Endo-IP) and provide proteomic and lipidomic snapshots of EEA1-positive endosomes in action. We identify recycling, regulatory and membrane fusion complexes, as well as candidate cargo, providing a proteomic landscape of early/sorting endosomes. To demonstrate the utility of the method, we combined Endo- and Lyso-IP with multiplexed targeted proteomics to provide a spatial digital snapshot of amyloid precursor protein (APP) processing by β and γ-Secretases, which produce amyloidogenic Aβ species, and quantify small molecule modulation of Secretase action on endosomes. We anticipate that the Endo-IP approach will facilitate systematic interrogation of processes that are coordinated on EEA1-positive endosomes.

The endolysosomal system is composed of a series of membrane-bound organelles that control much of the proteome flux within cells. Endosomes are dynamic organelles that are formed as a result of signaling events on the cytosolic face of the plasma membrane (PM), leading to vesicle budding from the PM through AP2-clathrin-dependent and independent processes[1–3]. Recently budded and uncoated endosomes coalesce into RAB5 GTPase-positive vesicles referred to as "sorting endosomes". These organelles receive additional cargo and regulatory proteins through fusion with other endosomes and transport vesicles, while also directing cargo towards distinct compartments such as RAB11-positive recycling endosomes[3–5]. Sorting endosomes also undergo maturation to form lysosomes, a major degradative compartment within cells. Endosome maturation is accompanied by loss of RAB5 and acquisition of RAB7 on late endosomes, as well as trafficking of macromolecules that support the production of functional acidified lysosomes (e.g. resident hydrolytic

[1]Department of Cell Biology, Blavatnik Institute, Harvard Medical School, Boston, MA 02115, USA. [2]Aligning Science Across Parkinson's (ASAP) Collaborative Research Network, Chevy Chase, MD 20815, USA. [3]Department of Biomolecular Chemistry, University of Wisconsin–Madison, Madison, WI 53706, USA. [4]Morgridge Institute for Research, Madison, WI 53715, USA. [5]Department of Chemistry, University of Wisconsin–Madison, Madison, WI 53706, USA. [6]Present address: Department of Dental Science, School of Dentistry and Dental Research Institute, Seoul National University, Seoul 03080, Republic of Korea. [7]Present address: Casma Therapeutics, Cambridge, MA 02139, USA. [8]These authors contributed equally: Frances V. Hundley, Qing Yu. ✉e-mail: wade_harper@hms.harvard.edu

enzymes)[6,7]. PM proteins destined for the degradative pathway are sorted into intra-lumenal vesicles characteristic of multi-vesicular bodies (MVBs) via the ESCRT system within late endosomes, and are ultimately degraded upon maturation to the lysosome. As such, early/sorting endosomes can be considered as a continuum of states with specific "domains" on an individual organelle representing intermediates in the conversion from an early endosome to a late endosome. Lysosomes are also critical for degradation of cytosolic proteins and organelles through the process of autophagy, wherein cargo-laden autophagosomes fuse with lysosomes to facilitate cargo degradation[8]. Finally, the endolysosomal system serves important signaling functions in cells, including both nutrient sensing via the mTOR system within lysosomes and proteolytic processing events that are coupled with endocytosis of PM proteins[9,10].

Our understanding of lysosomal function has been advanced through the ability to rapidly isolate intact organelles via immuno-precipitation (IP) using a resident integral lysosomal membrane protein TMEM192, referred to as *Lyso-IP*[11]. However, to our knowledge an analogous system has not been developed for early/sorting endosomes. To address this limitation, we developed a method that allows the selective affinity isolation of EEA1-positive early/sorting endosomes (termed *Endo-IP*) directly from tissue culture cell lysates, which when paired with Lyso-IP, allows spatial analysis of early and late compartments within the endolysosomal system. EEA1 is recruited to newly uncoated early endosomes and facilitates fusion with sorting endosomes, with which it also associates[3,12-14]. The array of proteins we find associated with early/sorting endosomes includes many factors known to function in maturation and sorting, as well as candidate cargo, and is distinguished from lysosomes by the absence of many lysosomal degradative enzymes. We demonstrate that endocytosed cargo can be detected by Endo-IP within minutes of exposure to cells, indicating that the approach can be used to dynamically examine early steps in the process.

To demonstrate the utility of the Endo-IP approach for spatial analysis of endosomes, we focused on juxta- and intra-membrane proteolytic processing of the Alzheimer's disease-associated protein APP to amyloid-forming Aβ species. APP is a single-pass transmembrane (TM) protein with multiple extracellular domains and a short C-terminal tail[15]. APP processing is complex and is thought to occur in multiple membrane compartments through pathways that are dictated, in part, by the trafficking the proteolytic enzymes (α-Secretase, the major β-Secretase BACE1, and γ-Secretase) themselves[16]. This complexity has led to multiple, and in some cases conflicting, models that describe where and how APP is processed into Aβ. Historically, APP was thought to be trafficked to the plasma membrane via the conventional secretion system where it could undergo either cleavage by α-Secretase to release its extracellular domain in the non-amyloidogenic pathway or undergo clathrin-dependent endocytosis and subsequently be processed to generate Aβ amyloid species[17-19]. However, recent work suggests that APP can be trafficked directly from the Golgi to EEA1-positive early endosomes via an AP-4-dependent process, and that this pathway may be the dominant trafficking route in some cell types[20,21]. There is consensus that processing of APP by the major β-Secretase BACE1 to generate the CTFβ product—a 99-residue fragment containing the TM and C-terminal domain—occurs primarily in an early endosomal compartment[16]. This reflects the fact that BACE1 and APP are jointly endocytosed[22,23] and that BACE1 activity is increased by the acidic environment of the endosome[24]. In addition, BACE1 is rapidly sorted into recycling endosomes and is therefore not present in the same compartment during subsequent APP trafficking through the endolysosomal system. In contrast, the cellular location(s) for further processing of CTFβ by PSEN1 or 2—the catalytic subunits of the intramembrane protease γ-Secretase[10]—to generate Aβ peptides are less well defined. On one hand, recent studies using induced neurons with PSEN1 or APP mutations revealed enlarged RAB5-positive endosomes whose

formation was BACE1-dependent and correlated with the abundance of CTFβ in the organelle, consistent with CTFβ processing by BACE1 in the early endosome[25]. On the other hand, studies in cancer cell lines indicated selective enrichment of PSEN2 in lysosomes where it was proposed to process CTFβ to produce Aβ[26]. Alternatively, γ-Secretase is also trafficked through the Golgi to the plasma membrane and defects in trafficking of APP from the Golgi to the endosome increased the level of extracellular Aβ, leading to the hypothesis that BACE1 and γ-Secretase can act on APP to generate Aβ when the residence time within the Golgi is extended[16]. Still other models have suggested that APP is trafficked from the Golgi directly to the lysosome where it is processed[27]. In neuronal dendrites, APP and BACE1 are thought to co-exist specifically in recycling endosomes in response to synaptic activity, and this association correlates with the appearance of CTFβ[28].

A limitation of previous studies examining APP/Aβ processing is the near universal analysis of APP/Aβ products using immunological detection in culture media or whole-cell extracts, thereby obscuring an understanding of the spatial accumulation of processed products. To address these limitations, we combined Endo-IP and Lyso-IP with a newly developed proteomics workflow that allows quantification of Aβ products derived from the action of BACE1 and γ-Secretase. This workflow, based on TOMAHAQ (triggered by offset, multiplexed, accurate-mass, high-resolution, and absolute quantification)[29,30], employs reference peptides designed to identify and quantify APP/Aβ products as "half-tryptic" peptides, thereby providing a digital spatial snapshot of APP/Aβ processing within individual organelles. Using this approach, we demonstrate that early/sorting endosomes contain substantial levels of Aβ cleavage products derived from both BACE1 and γ-secretase activities, and that these products are maintained in lysosomes. Moreover, we demonstrate that this approach can be used to quantify the effect of modulators of γ-Secretase activity at the level of individual organelles.

## Results

### Evaluation of candidate endosomal proteins for Endo-IP

We sought to develop an approach analogous to Lyso-IP for purification of early endosomes directly from cell extracts. First, we adopted a previously reported targeting approach[31] to create HEK293 cells (referred to throughout as 293) wherein endogenous TMEM192 was tagged with a 3xHA epitope (referred to as 293[L] for Lyso-IP cells) (Supplementary Fig. 1a; see "Methods"). We then stably expressed FLAG-tagged proteins known to associate with early/sorting endosomes (RAB5A, EEA1), recycling endosomes (RAB11A), or both (TFR1) in 293[L] cells (Supplementary Fig. 1b), and profiled the enrichment of known endosomal proteins after α-FLAG IP compared with 293[L] cells lacking a tagged protein using Tandem Mass Tagging (TMT)-based quantitative proteomics. Cell lysis was performed without the use of detergents to maintain organelle integrity. The strongest enrichment of endosomal proteins, as annotated in Itzhak et al.[32], was found with FLAG-EEA1 (Fig. 1a; Supplementary Fig. 1c; Supplementary Data 1). EEA1 contains a C-terminal FYVE domain that associates with PI3P on recently uncoated endosomes and an N-terminal domain that binds to RAB5 on sorting endosomes, to facilitate endosomal fusion[13,14]. RAB5-positive sorting endosomes also contain significant levels of PI3P in particular membrane "domains" which can also associate with EEA1. However, during maturation, PI3P on sorting endosomes is converted to PI(3,5)P$_2$, resulting ultimately in complete loss of EEA1 and replacement of RAB5 by RAB7 to generate late endosomes[33]. We therefore reasoned that PI3P abundance on the early/sorting endosome could serves as a timer for capture by EEA1 that would provide specificity for early and/or sorting endosomes.

### Evaluation of EEA1 for Endo-IP

To prepare a homogeneous population of the cells expressing near-endogenous level of FLAG-EEA1, we stably expressed FLAG-EEA1 in

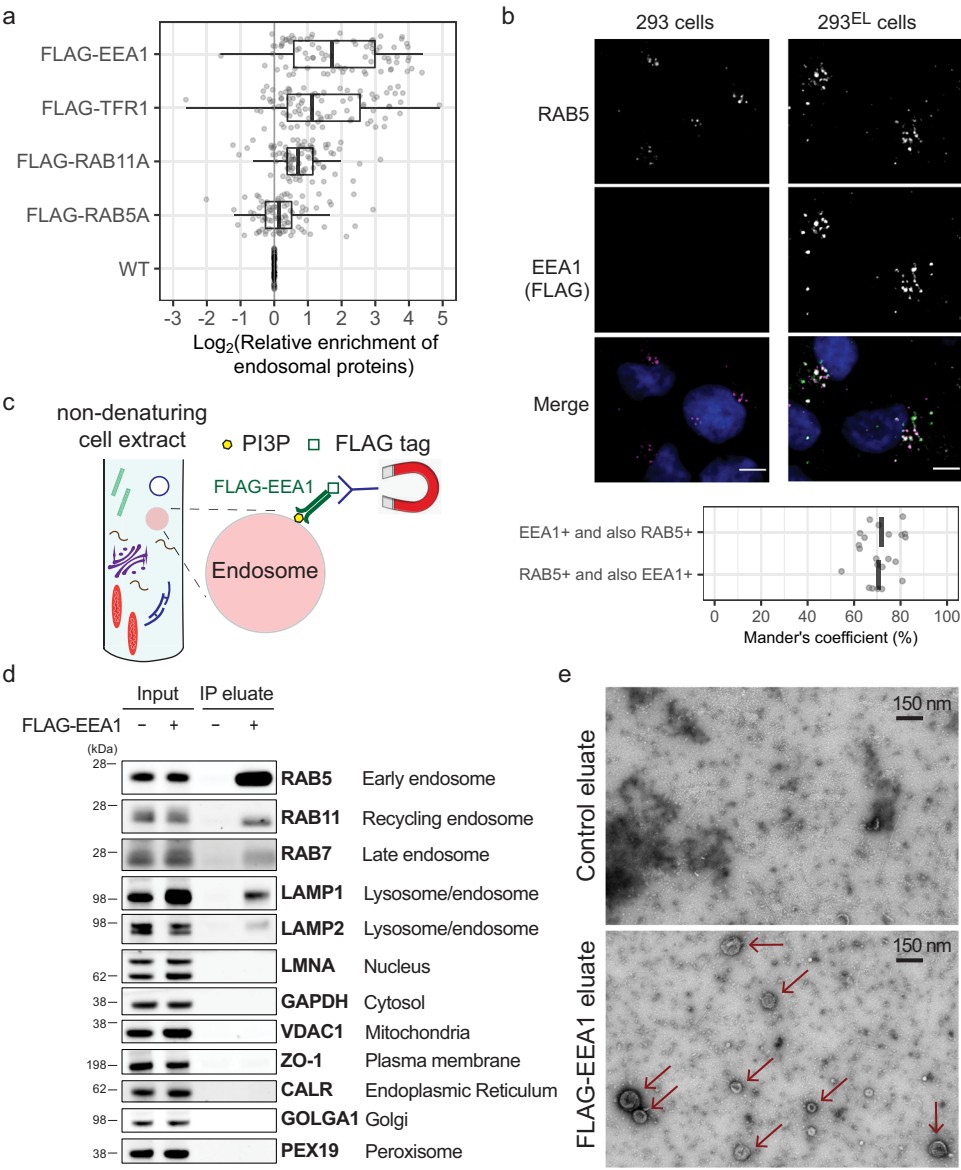

**Fig. 1 | Identification of EEA1 as a candidate affinity reagent for early endosome purification. a** The indicated FLAG-tagged proteins were stably expressed in 293[L] cells and non-detergent extracts subjected to immunoprecipitation with α-FLAG antibodies prior to proteomic analysis of trypsinized peptides using 10-plex TMT-based proteomics which included one replicate each of immunoprecipitates and whole-cell extracts in the 10-plex. The enrichment (Log$_2$FC) of endosomal proteins[32] (gray circles) relative to control 293[L] cells is shown, with FLAG-EEA1 displaying the greatest enrichment of proteins known to localize to endosomes. Left border, interior line, and right border in the box plot represent the 1st quartile, median, and 3rd quartile, respectively. **b** 293[EL] cells (clone 33) were subjected with immunofluorescence using α-FLAG (green, to detect FLAG-EEA1) and α-RAB5 (magenta) antibodies, and nuclei were stained with Hoechst 33342 (blue). Image

analysis of 11 cells indicates that the Mander's coefficient is ~0.7 for both the overlap of FLAG puncta with RAB5 and RAB5 puncta with FLAG. **c** Scheme depicting affinity purification of PI3P-positive early endosomes using FLAG-EEA1 in conjunction with α-FLAG antibodies immobilized on magnetic beads. **d** Control 293 cells or 293[EL] cells (clone 33) were lysed in non-denaturing buffer prior to either direct analysis of extracts by immunoblotting or were subjected to α-FLAG magnetic bead capture followed by immunoblotting with the indicated antibodies. Loaded amounts are equivalent to 0.06% and 6% of the input and IP eluate, respectively. **e** α-FLAG immunoprecipitates from either control 293 cells or 293[EL] cells (clone 33) were released from the affinity resin by FLAG peptide and then analyzed by transmission EM. Arrowheads indicate vesicular structures present in cells expressing FLAG-EEA1 but not control cells. Scale bar, 150 nm.

293[L] cells, thereby generating 293[EL] cells and selected two independent clonal population expressing FLAG-EEA1 at ~3.0-fold higher levels than the endogenous protein (Supplementary Fig. 1d, https://doi.org/10.5281/zenodo.7177916). Consistent with association of FLAG-EEA1 with early/sorting endosomes, FLAG-EEA1 was preferentially found in cytosolic puncta as assessed by imaging, the majority (~70%) of which were positive for staining with α-RAB5 (Fig. 1b). Likewise, ~70% of RAB5-positive puncta were positive for FLAG-EEA1 (Fig. 1b). As a control, we determined expression of FLAG-EEA1 in the context of TMEM192[HA] and found it had little effect on the abundance of ~6000 proteins based on total proteome analysis using TMT when compared

with parental 293 cells (Supplementary Fig. 1e, f; Supplementary Data 2) (Correlation coefficient > 0.97), indicating that FLAG-EEA1 expression was not detrimental to cells. Consistent with FLAG-EEA1 preferentially associating with early/sorting endosomes, FLAG-EEA1 immune complexes were highly enriched for RAB5 based on immunoblotting (Fig. 1c, d). In addition, we detected LAMP1, which is present on a subset of early endosomes, as well as RAB11 and RAB7, as would be expected for sorting endosomes that are actively involved in formation of recycling or maturation towards late endosomes, respectively (Fig. 1d). However, proteins associated with the nucleus (LMNA), cytosol (GAPDH), mitochondria (VDAC1), ER (CALR), or Golgi

(GOLGA1) were not detected (Fig. 1d). Like 293 cells, FLAG-EEA1-expressing cells display transferrin (TF) uptake at both 5 and 15 min post TF treatment (Supplementary Fig. 1g). Consistent with isolation of intact organelles, transmission electron microscopy of FLAG-EEA1 associated vesicles released from the affinity matrix revealed the presence of particles with a median size of 110 nm not observed in α-FLAG immune complexes from control cells (Fig. 1e and Supplementary Fig. 1h). The 25th percentile of particles had a size of ~85 nm, indicating that FLAG-EEA1 can recover newly uncoated early endosomes, while also allowing the identification of larger particles up to ~240 nm that could reflect more mature or sorting endosomes (Supplementary Fig. 1h).

### Proteomic landscape of the EEA1-positive endosomes

Our understanding of endosomal proteome composition is based largely on gradient-purified vesicles, which appears to represent a continuum of endosomal states based on the identification of proteins linked with early, late, and recycling endosomes[32]. As an unbiased assessment of the performance of FLAG-EEA1 for isolation of early/sorting endosomes, which make up ~0.9% of the total cellular proteome, we performed quadruplicate α-FLAG IPs in both control 293 and 293[EL] cells and analyzed associated proteins using 8-plex TMT (Fig. 2a and Supplementary Data 3). In total, 316 proteins displayed enrichment in the FLAG-EEA1 immune complex (Log$_2$ FC > 1.0, $p$ value <0.02) (Fig. 2b). FLAG-EEA1 immunoprecipitates were dramatically enriched in proteins with links to endosomes and the plasma membrane, but were not enriched in a variety of other organelles, including ER, Golgi, peroxisomes, and mitochondria (Fig. 2c). Indeed, >219 of the enriched proteins have known association with endosomes, the endolysosmal system, vesicle fusion, or are candidate endocytic cargo by virtue of PM association (Fig. 2b–d, Supplementary Data 3). Many of these proteins are organized into functional modules or classes in Fig. 2d. In addition to proteins thought to be "resident" endosomal proteins (e.g. LAMP1, TMEM9/9B), we identified numerous members of multi-subunit protein complexes associated with endosomes, including the V-ATPase responsible for acidification of the endolysosomal system; ESCRT-III, which functions in sorting of proteins into MVBs; the HOPS complex, which promotes exchange of RAB5 by RAB7 during maturation; the retromer complex and its specific sorting nexins; and components of the AP1 complex involved in formation of recycling endosomes (Fig. 2d; see below). We detected multiple R-snare VAMP proteins as well as multiple sub-classes of T-snares, fusion proteins, and regulatory components, many of which have been shown to be involved in aspects of endosomal membrane fusion (Fig. 2d)[34]. Although relatively poorly understood, we also identified three RUN and FYVE (RUFY) proteins, which are known to localize on endosomes and coordinate multiple RAB GTPase circuits[35]. These RUFY proteins are known to interact physically or be in proximity with EEA1[36], as well as components of the endosomal membrane (LAMTOR1) and the snare system (STX7 and WDFY1) (Fig. 2d)[36].

We compared proteins enriched by Endo-IP with those identified in HeLa cells by gradient purification followed by proteomics[32]. In total, 93 proteins were found in common (Fig. 2e, Supplementary Data 3), including proteins in most functional classes identified by Endo-IP (marked with asterisks in Fig. 2d). In total, 63 known endosome-related proteins were identified by Endo-IP that were not seen in gradient fractionation whereas 54 endosome-related proteins were found with gradient fractionated samples not detected with a Log$_2$FC > 1.0 with Endo-IP (Supplementary Data 3). Among the selective Endo-IP proteins were various sorting nexin fusion machinery, several RAB network proteins, and a subset of sorting receptors (Fig. 2d). Interestingly, among the proteins selectively identified in gradient purified endosomes were proteins more closely linked with late endosomes, including components of the AP3 complex which delivers cargo from the Golgi to the late endosome, CCZ1-MON1B which acts as a GEF for

RAB7 on late endosomes, and RAB11 which marks recycling endosomes (Supplementary Data 3). The strong enrichment of numerous proteins linked with endosomal functions indicates that FLAG-EEA1 can be used as a tool for "Endo-IP", and this together with Lyso-IP from the same cells via the dual tagging strategy (Supplementary Fig. 1a, d) allows spatial and temporal analysis of the endocytic system.

During maturation, lysosomes accumulate a number of components that promote formation of a functional degradative organelle, including lumenal degradative enzymes (cathepsins, nucleases, lipid metabolism enzymes). To further validate the ability of Endo-IP to distinguish early/sorting endosomal and lysosomal compartments, we performed α-HA-based Lyso-IPs[11] on 293 and 293[EL] cells followed by immunoblotting (Supplementary Fig. 2b) or 8-Plex TMT proteomics (Supplementary Fig. 2c, d and Supplementary Data 4). As expected, Lyso-IPs were highly enriched in annotated lysosomal proteins, including LAMP1 as seen by immunoblotting, but not proteins associated with other organelles (Supplementary Fig. 2b, d)[32]. In total, 91 proteins were found to be enriched in the Lyso-IP relative to untagged cells (Log$_2$ FC > 1.0, $p$ value <0.02), and 67 of these are known lysosomal proteins (Supplementary Fig. 2c, e and Supplementary Data 4). Eight additional lysosomal proteins were present with sub-threshold $p$ values or with Log$_2$ FC > 0.8 (marked with asterisks in Supplementary Fig. 2e). Enriched proteins included 41 lumenal enzymes and 24 proteins associated with the lysosomal membrane (Supplementary Fig. 2e). We detected 20 proteins in common between the Endo-IP and the Lyso-IP, including components of the BLOC and LAMTOR complexes. Importantly, however, only 7 of the 41 canonical lysosomal lumenal enzymes typical of mature lysosomes were observed in common with the Endo-IP (Supplementary Fig. 2e). Conversely, essentially all of the endosomal sorting and membrane fusion proteins were selectively found with the Endo-IP, including RUFY, WDFY, RABEP, sorting receptors, the phosphoinositide phosphatase VAC14, PI3P binding components, and RABs 4, 5 and 14 (Supplementary Fig. 2e). These results indicate selective enrichment of distinct compartments within the endolysosomal system using Endo- and Lyso-IP approaches.

### Endo-IP facilitates dynamic capture of endosomal cargo

Ligand binding to cell surface receptors is frequently associated with rapid internalization of endosomes, which become associated with EEA1, typically within minutes. Previous studies have examined endosomal content 20 min post-stimulation using conventional gradient centrifugation of organelles[32]. To examine the ability of Endo-IP to detect rapid endocytic events, we focused on TF, as it is detected in Endo-IPs at steady-state (Fig. 2b, e) and is also rapidly internalized when added to serum-starved cells as described above. 293[EL] cells were deprived of serum for 1 h, and subsequently supplemented with TF (25 μg/ml) in serum-free media (Fig. 3a). Cells were then harvested at 0, 5, 15, and 30 min followed by lysis and Endo-IP, and purified protein subjected to immunoblotting (Fig. 3b). While TF was not detected in EEA1-positive endosomes in serum-starved cells, it was readily detected in samples from cells harvested at 5 min post-stimulation, and was maintained throughout the time course (Fig. 3b). As a further approach to probe dynamics of endocytosis, we performed an analogous experiment wherein cultures of 293 or 293[EL] cells (in biological duplicate) were subjected to serum starvation and TF treatment (Fig. 3a). EEA1-positive endosomes from 293[EL] cells, but not 293 cells, displayed dramatic enrichment of TF at 5 and 15 min based on both immunoblotting (Supplementary Fig. 3a, b) and correlation plots of Log$_2$FC of Endo-IPs from 293[EL] or 293 controls cells determined by proteomics (Fig. 3c and Supplementary Data 5). The dramatic increase in TF in early endosomes is also seen in bar graphs of TF signal-to-noise at both 5 and 15 min in replicate analyses (Fig. 3d and Supplementary Data 5).

As an additional test of the utility of the method for quantifying changes in endosomal contents, we performed Endo-IPs on 293[EL] cells

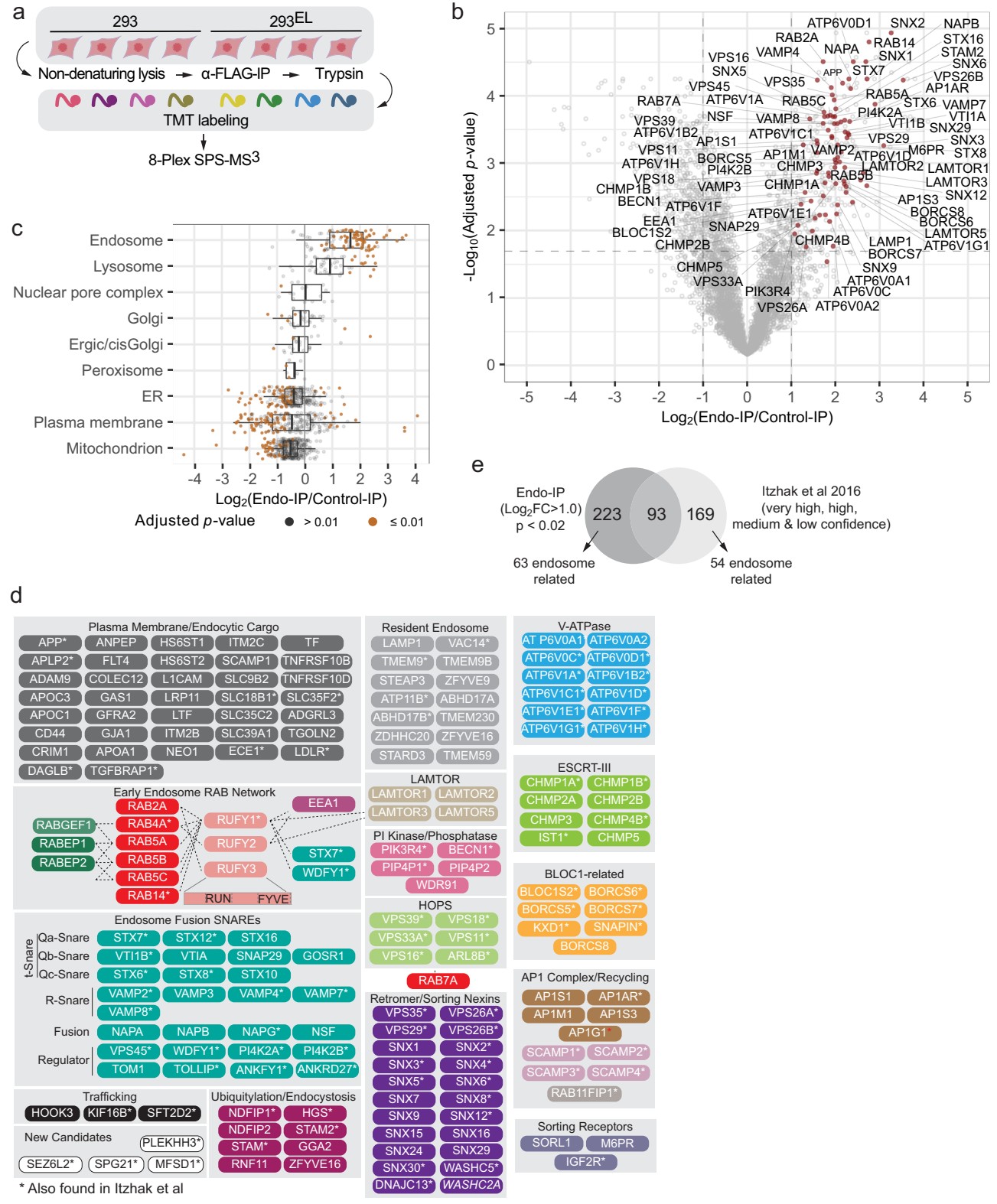

with endocytosis blocked using a hydroxylated version of the small molecule "Dynasore"[37] (referred to as Hydroxy Dynasore or Dyngo™4a[38]). Hydroxy Dynasore−an inhibitor of DNM1/2 required for scission of clathrin-coated pits from the PM−can also inhibit DNM-independent endocytosis[39]. As expected, treatment of 293[EL] cells with Hydroxy Dynasore (3 h) resulted in maintenance of a subset of EEA1 and RAB5-positive vesicles, as assessed by immunofluorescence, although vesicles in treated cells were swollen (Supplementary Fig. 3c).

The fraction of EEA1 and RAB5 vesicles with Hydroxy Dynasore treatment was similar to that seen in control cells (Mander's coefficient -0.7) (Supplementary Fig. 3c). We found that endosomes from Hydroxy Dynasore-treated cells maintain association with the majority of resident and regulatory endosomal proteins (Log₂ FC < ±0.8), but were dramatically de-enriched in candidate endocytic cargo proteins (Log₂ FC < −1.0), including TFRC, VLDLR, LDLR, and LRP8 (Fig. 3e and Supplementary Data 5). Interestingly, among the most affected

**Fig. 2 | Landscape of the early/sorting endosome proteome. a** Schematic depicting an approach for detailed analysis of the early endosome proteome based on Endo-IP. Cells were employed in biological quadruplicate using 293 cells as a control ($n = 4$). Immune complexes were digested with trypsin, peptides labeled using 8-plex TMT and analyzed by mass spectrometry. **b** Volcano plot for quadruple Endo-IP associated proteins relative to control cells lacking tagged EEA1. Two-sided Student's $t$-test was performed and adjusted for multiple comparisons by two-stage Benjamini & Hochberg step-up procedure. The dashed lines indicate threshold of $Log_2FC > 1.0$ with $p$ value less than 0.02. Selected proteins linked physically or functionally with early endosomes are shown in red. This experiment employed 293[EL] cells (clone 33). **c** Box plots depicting the enrichment of various classes of proteins based on the annotation of Itzhak et al.[32] demonstrating that proteins assigned to endosomes and to a lesser extent, lysosomes are the most highly enriched, while ER, mitochondrial, and a subset of PM proteins are depleted. Left border, interior line, and right border in the box plot represent the 1st quartile, median, and 3rd quartile, respectively. Two-sided Student's $t$-test was performed and adjusted for multiple comparisons by two-stage Benjamini & Hochberg step-up

procedure. **d** Proteins significantly enriched in the Endo-IP are organized into functional modules. All the proteins shown were identified as being enriched with $Log_2FC > 1.0$ in the FLAG-EEA1 sample, with the exception of WASHC2A (indicated in italic) which had a $Log_2FC$ value of 0.99. Dotted lines for the early endosome RAB network indicate the presence of physical interactions or association based on proximity biotinylation experiments. Proteins indicated with an asterisk were found in common with the endosomal compartment proteome reported by Ihtzak et al.[32]. See Supplementary Data 3. **e** Comparison of proteins enriched in Endo-IP with the endosomal proteome identified by Ihtzak et al, 2016 using gradient fractionation[32]. Two-sided Student's $t$-test was performed and adjusted for multiple comparisons by two-stage Benjamini & Hochberg step-up procedure. Proteins (262) in the very high, high, medium, and low confidence intervals from HeLa cells[32] and 316 proteins from the Endo-IP from 293[EL] cells ($Log_2FC > 1.0$) were used, with the exception of proteins with a red asterisk which were just below the fold change cut-off. Proteins overlapping in the two data sets are indicated with asterisks in panel **d**.

proteins was MRC2, a C-type mannose endocytic lectin receptor that internalizes glycosylated ligands from the extracellular space[40]. Similarly, SORT1 which may function in part by scavenging extracellular proteins via endocytosis is also lost from endosomes upon DNM1/2 inhibition (Fig. 3e)[41]. Thus, the Endo-IP approach is capable of revealing dynamic changes in endosomal cargo capture initiated from the plasma membrane.

## Lipidomic snapshot of EEA1-positive endosomes

To establish a proof-of-concept that our enrichment method is compatible with downstream analytical technologies beyond proteomics, we next performed discovery lipidomics on vesicles isolated by triplicate Endo-IPs. Briefly, in this approach, which we have used widely for relative quantification of lipids in numerous biological systems[42–44], isolated lipids are separated and mass analyzed using liquid chromatography coupled to mass spectrometry by electrospray ionization. Lipids are identified using an accurate mass measurement along with tandem mass spectrometry. The mass-to-charge ($m/z$) peaks of the identified lipid species are then aligned across the various data files and relative abundance values calculated[42–44]. We note that while there are several methodologies to measure lipids, here we selected the discovery approach with relative quantification with the goal of directly comparing which lipid molecules are enriched with the Endo-IP vs. control-IP—i.e., comparison within the same study relative quantification is highly appropriate. Further, because these samples result from an IP, very little material is obtained making it challenging to determine total lipid mass in each sample, which is essential for absolute concentration calculations.

In total, we identified 276 individual lipid species present in either Endo-IP or control IPs from 293[EL] cells in biological triplicate, which could be placed into 20 lipid classes (Supplementary Data 6). We then identified individual lipid species that were enriched in the Endo-IP ($log_2 FC > 2.0$) and that also passed a strict $p$ value cut-off of 0.01 for replicate analyses (Supplementary Fig. 3d). Endo-IPs were enriched in phosphatidylserine (PS), sphingomyelin (SM), HexCer-NS, plasmenyl/plasmanyl (ether-containing) forms of phosphatidylethanolamine (PE) and phosphatidylcholine (PC), as well as non-ether PC derivatives (lipids with $p$ values < 0.01 indicated in orange and >0.01 indicated in gray) (Fig. 3f and Supplementary Fig. 3d and Supplementary Data 6). This set of lipids fits well with lipids known to be present in the plasma membrane[45]. Interestingly, PC as well as plasmanyl-PC gave a bi-modal distribution of significantly enriched species. Enriched species were almost completely derived from saturated or mono-unsaturated fatty acids while de-enriched PC species were populated by primarily poly-unsaturated fatty acids (Supplementary Fig. 3e). Previous studies indicate that poly-unsaturated lipids are asymmetrically localized on the inner leaflet of the plasma membrane of red blood cells while

unsaturated or mono-unsaturated lipids are predominantly exoplasmic[45]. Unsaturated, and particularly poly-unsaturated fatty acids, may generate more fluid membrane domains and have been implicated in regions where membrane fission occurs, whereas saturated fatty acids are associated with thicker and less fluid membranes[3,46]. Notably a common lysosomal lipid bis(monoacylglycerol)phosphate (BMP) was not identified in the original analysis. To further explore abundance of BMPs in endosomes, we analyzed Endo-IP and Lyso-IP samples in parallel using a modified protocol (see "Methods"). We found that BMPs were identified and significantly enriched in Lyso-IP, however, were substantially less enriched in EEA1-positive endosomes (Supplementary Fig. 3f and Supplementary Data 6). This is consistent with prior studies indicating that BMP is localized primarily in late endosomes and lysosomes[47], and further explains why BMPs were not identified in the original lipidomics analysis. Moreover, as expected, lipids enriched in mitochondria (cardiolipin, CP) as well as non-ether forms of PE are de-enriched in early endosomes (Fig. 3f). Future experiments that establish the precise concentrations the lipids in endosomes would extend this work in several ways, for example, allowing direct comparison to other studies, enabling molecular modeling where precise concentrations are needed, and aiding in the establishment of in vitro endosome models.

## A TOMAHAQ toolkit for quantifying APP/Aβ processing

Having developed Endo-IP, we next sought to examine its utility for analysis of regulatory events associated with endosomes. Among the candidate cargo enriched in Endo-IPs was the APP amyloid precursor protein (Fig. 2b, d). APP is composed of a large multi-domain extracellular region, a single transmembrane segment, and a short C-terminal tail, and is thought to be proteolytically processed within the endolysosomal system (Fig. 4a)[15,48]. Delivery of APP to early/sorting endosomes by either endocytosis or direct delivery from the Golgi via AP-4 results in its extracellular domain being present within the endosomal lumen (Fig. 4a)[16,20]. While early and recycling endosomes likely represent major sites of APP processing by the major β-Secretase BACE1 to generate a 99-residue transmembrane segment-containing CTFβ product[16,22,23,28,49], the location of cleavage by γ-Secretase is controversial with some studies pointing to processing selectively in the lysosome while others suggest endosomal or Golgi processing sites[17,25,26] (Fig. 4a). The catalytic subunits of γ-Secretase, PSEN1 and PSEN2[10], cleave CTFβ at the membrane/cytosol boundary via successive release of tri- and tetra-peptides to form Aβ40 and 42, and in some cases is further processed to Aβ37 and 38 (Fig. 4a)[50–52].

To quantify processed APP in endolysosomal compartments, we reasoned that "half-tryptic" peptides—representing endogenous cleavage by BACE1 or γ-Secretase and subsequent cleavage of isolated compartments by trypsin in vitro—would generate a unique species for

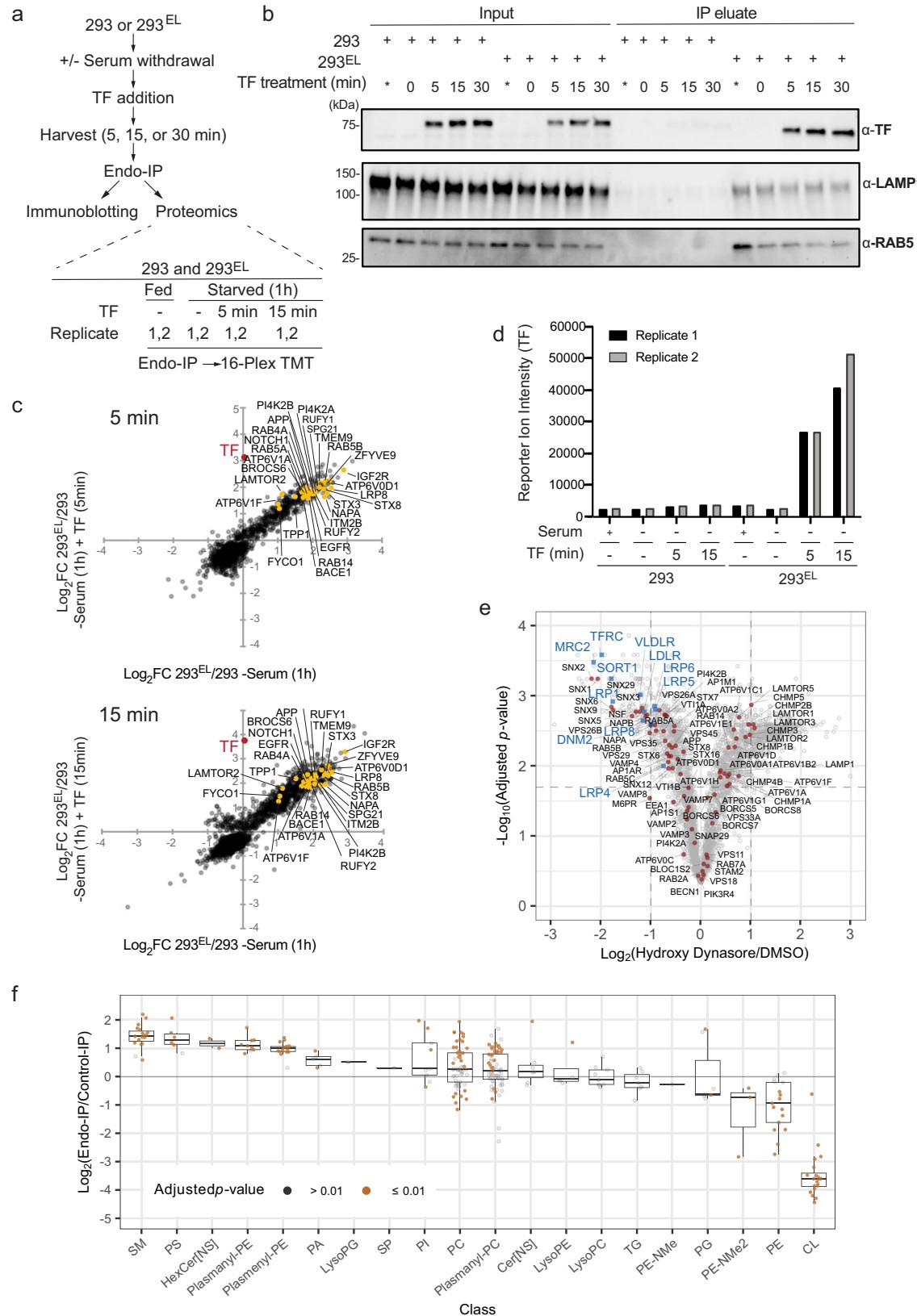

analysis using TOMAHAQ-based targeted proteomics (Fig. 4b, c and Supplementary Data 7)[29]. To initially test this approach, we digested synthetic Aβ40, Aβ42 and Aβ43 peptides with trypsin, labeled the products with TMT-126, mixed, and then combined with synthetic reference half-tryptic Aβ trigger peptides representing the products of cleavage by BACE1 and γ-Secretase that were labeled with TMTsh

(super-heavy) (Fig. 4b and Supplementary Data 8). A dilution series of the products over 5 orders of magnitude were then mixed with TMT-unlabeled cell extracts from 293 cells (see "Methods") to simulate detection within a complex sample prior to six independent TOMA-HAQ analyses. For TOMAHAQ, TMTsh-reference half-tryptic peptides were used in real time to trigger quantification of TMT reporter ions

**Fig. 3 | Analysis of Endo-IP for capture of dynamic cargo and lipidome analysis. a** Scheme depicting the analysis of TF (transferrin) uptake by Endo-IP. **b** 293 or 293^EL cells (clone 33) were serum starved for 1 h prior to addition of serum-free media with or without 25 µg/ml holo-transferrin (TF). At the indicated times, cells were harvested and subjected to Endo-IP. Control samples not subjected to serum starvation (*). For input, 0.5% of total was subjected to immunoblotting, whereas 6% of the elution was analyzed. Two separate experiments were performed, each with n = 1. **c** Proteomic enrichment of Endo-IPs from the experiment outlined in panel a with 5 and 15 min of TF treatment were plotted as a correlation plot with the average from biological duplicates (n = 2). Enrichment of TF in 5 and 15 min samples is indicated in red. Enrichment of a set of endosomally localized proteins in 293^EL cells compared with the 293 cell controls are indicated. Immunoblots for this experiment are shown in Supplementary Fig. 3a, b. Also see Supplementary Data 5. **d** Reporter ion signal-to-noise ratios for TF (biological duplicate measurements) from the experiment in panel c are shown in the bar graphs. Also see Supplementary Data 5. **e** 293^EL cells (clone 33) were either left untreated (DMSO) or were

treated with the DNM1/2 inhibitor Hydroxy Dynasore in triplicate (n = 3) for 3 h to disrupt endocytosis. Non-detergent extracts were subjected to immunoprecipitation prior to analysis by mass spectrometry. The volcano plot shows that the majority of endosomal proteins indicated as red dots are largely unchanged, but several endocytic cargoes (blue squares) are substantially reduced (Log$_2$FC < −1.0). Two-sided Student's t-test was performed and adjusted for multiple comparisons by two-stage Benjamini & Hochberg step-up procedure. **f** Triplicates of Endo-IP and control immune complexes from 293^EL cells (clone 33) were analyzed using the Lipidex platform[42] to identify major enriched lipid species. The abundance of the indicated lipid classes compared with control immune complexes are shown in the box plot (n = 3). Lower border, interior line, and upper border in the box plot represent the 1st quartile, median, and 3rd quartile, respectively. Two-sided Student's t-test was performed and adjusted for multiple comparisons by two-stage Benjamini & Hochberg step-up procedure. Lipid species with p values less than 0.01 are indicated in orange while those with p values greater than 0.01 are indicated in gray.

from co-migrating multiplexed target Aβ peptides present at a known offset using the Tomahto application programming interface[30] (Fig. 4c, Supplementary Fig. 4a–d). MS² analysis of target peptides, as shown for Aβ40 (Fig. 4c), enables the selection of interference-free b- or y-type fragment ions as precursors (synchronous precursor selection [SPS]) for an MS³ spectrum with reporter ion quantification. Half-tryptic peptides for cleavage by γ-Secretase were detectable at less than 200 amol, while the half-tryptic peptide for cleavage of Aβ peptides by BACE1 could be detected at 600 amol (Fig. 4d).

We next prepared a series of synthetic reference half-tryptic peptides with the goal of monitoring APP processing by BACE1 and γ-Secretase, including Aβ34-Aβ43 species (Fig. 4e and Supplementary Data 7). We also prepared peptides derived from extracellular (E1, KPI, E2, and juxtamembrane), and C-terminal domains (including phosphorylated and unphosphorylated forms of the CTD) (Fig. 4e and Supplementary Data 7) (see "Methods"). The chromatographic and fragmentation performance of TMTsh-trigger peptides was extensively characterized and optimized, allowing for identification and quantification of up to 32 peptides in a single multiplexed experiment (Fig. 4f and Supplementary Fig. 4a–d and Supplementary Data 7), as described below. Of note, APP has multiple isoforms derived from alternative splicing. Our reference peptide collection includes diagnostic peptides for the APP770, APP751, and APP695 isoforms (Fig. 4e).

**Coupling APP/Aβ-TOMAHAQ with endolysosomal enrichment**
We then sought to couple APP/Aβ-TOMAHAQ with endosome or lysosome enrichment. As we were unable to detect appreciable APP processing to CTFα/β with WT APP in 293 cells (Supplementary Fig. 5a, lane 1), we employed a mutant replacement strategy to enhance processing (Fig. 5a). We first used CRISPR/Cas9 to delete APP in 293^EL cells (Fig. 5a and Supplementary Fig. 5b). To mimic APP processing seen in familial AD, we then stably expressed a compound patient mutant—APP^K651N/M652L/T700N—at levels sixfold above that found basally in 293 cells (Supplementary Fig. 5a, https://doi.org/10.5281/zenodo.7177916). We employed the APP751 isoform as this is the isoform expressed in many non-neuronal cells such as 293 cells. Mutation of K651N/M652L (which we refer to as the Swedish (Sw) mutant) promotes cleavage by BACE1 while the T700N mutant enhances cleavage by γ-Secretase leading to increased levels of Aβ42[19]. For simplicity, we refer to 293^EL APP^−/− cells stably expressing APP^Sw/T700N as 293^EL-APP* (Fig. 5a). Endo-IPs from 293^EL-APP* cells contained primarily more slowly migrating forms of APP as examined by immunoblotting, while in contrast, Lyso-IPs from these cells contained both more slowly and more rapidly migrating forms (Supplementary Fig. 5c). The level of APP* expression was not sufficient to induce ATF4 expression as a marker of the integrated stress response (Supplementary Fig. 5d).

We then developed a workflow to detect APP and Aβ within endosomes and lysosomes using TOMAHAQ proteomics[29] (Fig. 5b). To initially examine the ability to detect Aβ cleavage products directly within Endo- and Lyso-IPs, we analyzed samples from biological duplicate 293^EL-APP* and biological duplicate 293^EL-APP^−/− cells in independent 4-plex TOMAHAQ experiments (Fig. 5b, c and Supplementary Fig. 5e, f). Within both Endo- and Lyso-IPs, we detected a subset of peptides derived from extracellular, transmembrane, and C-terminal domains, but half-tryptic peptides derived from various Aβ species were either not detected or were present with very low signal-to-noise (S/N) ratios, with the noise defined by MS³ reporter ion intensity present in APP^−/− cells (Fig. 5c, d and Supplementary Fig. 5e, f and Supplementary Data 9).

We reasoned that highly abundant proteins within organelle IPs could suppress detection of low abundance and very hydrophobic Aβ-derived peptides. We therefore developed an additional purification step involving low molecular weight (LMW) filtration (Fig. 5b), which we predicted would allow for the removal of the majority of proteins while maintaining the comparatively small-sized solubilized Aβ-related peptides (~4KDa) present in Endo and Lyso-IPs. To test this and to determine the optimal pore size of the filter, we solubilized synthetic Aβ38 in 8 M urea/0.5% NP-40 (which we found also solubilizes Aβ42/43-oligomers, Supplementary Fig. 5g), passed samples through 10, 30 or 50 kDa cut-off filters and then compared input, retentate, and filtrate samples for the presence of Aβ38 (Fig. 5e). The majority of Aβ38 was found in the filtrate with the 50 kDa filter, but was substantially excluded from the filtrate with 10 and 30 kDa filters. When applied to Endo- and Lyso-IP samples, we found that the majority of full-length APP (APP^FL), as well as CTFα/β, remained in the retentate (Fig. 5f and Supplementary Fig. 5c). Importantly, analysis of filtrate samples derived from Endo- and Lyso-IPs by TOMAHAQ revealed a dramatic increase in the S/N ratios for the vast majority of half-tryptic γ-Secretase products of APP, when compared with Endo and Lyso-IPs examined directly, including Aβ34, 37, 38, 39, 40 and 42 (Fig. 5g and Supplementary Fig. 5e and Supplementary Data 9). Thus, Endo- and Lyso-IP in combination with LMW filtration provides a potential route for specific detection of Aβ peptides occurring as a result of γ-Secretase activity. We did not reliably detect Aβ43 from either Endo- or Lyso-IP_LMW samples, despite our ability to quantify this peptide in the samples derived from in vitro cleavage (Fig. 4d). We refer to these enriched samples via Aβ filtration as Endo- or Lyso-IP_LMW (Fig. 5b).

The majority of unmodified non-Aβ peptides in APP are detected directly in Endo- or Lyso-IPs, albeit with varying signal-to-noise ratios (Supplementary Fig. 5f). Although the vast majority of APP^FL and CTFα/β remain in the retentate (Fig. 5f), the sensitivity of the TOMAHAQ method, coupled with removal of the many interfering proteins, nevertheless allows detection of several extracellular and C-terminal region peptides from small amounts of APP^FL/CTFα/β (or possibly

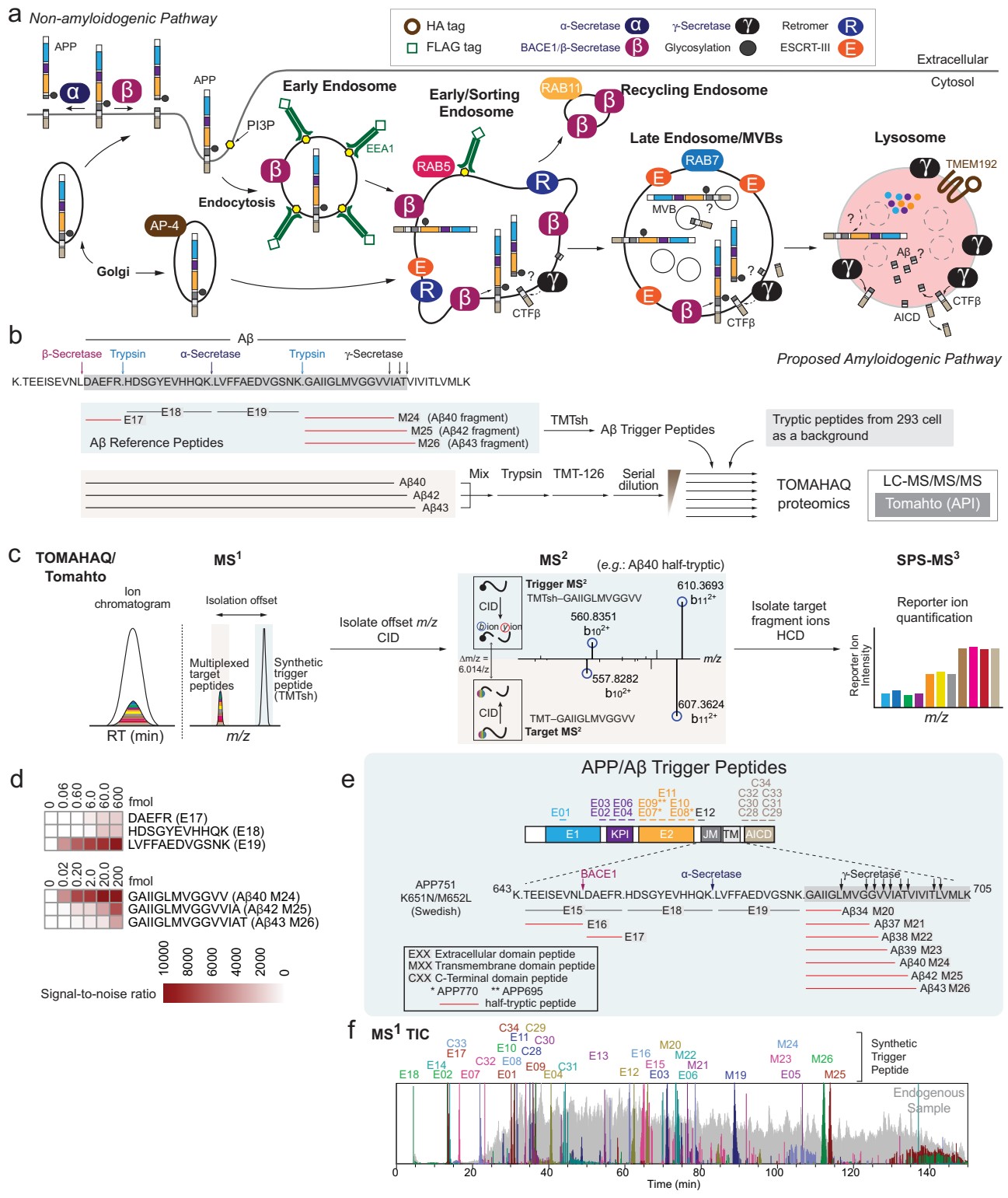

other proteolytic fragments) that may pass non-uniformly through the filter and are contained within the filtrate (Supplementary Fig. 5c, e, f). In particular, processing by BACE1 would release one half-tryptic peptide as part of sAPPβ into the organelle lumen while the other half-tryptic peptide would be tethered to the membrane as part of CTFβ (Fig. 4a). As such, we focused on the analysis of Aβ half-tryptic peptides derived from γ-Secretase cleavage using LMW filtration products and on half-tryptic peptide BACE1 cleavage products via direct analysis of organelle IPs, as described below.

## Digital snapshots of processed Aβ in endolysosomal compartments

BACE1 is known to be trafficked to the early endosome, and BACE1 ablation reduces the amyloidogenic pathway[53]. A previous study suggested that the γ-Secretase catalytic subunit PSEN2 is selectively present in lysosomes based on immunostaining, and therefore proposed that APP processing by γ-Secretase occurred specifically in lysosomes[26]. Detection of Aβ cleavage products within purified early endosomes led us therefore to examine the abundance of BACE1 and γ-

**Fig. 4 | A toolkit for analysis of APP processing in the endolysosomal system. a** Schematic overview of proposed APP processing events in the endolysosomal system. APP has an extracellular domain containing E1 (blue), KPI (purple), E2 (orange), juxtamembrane (dark gray), a transmembrane domain (light gray), and a C-terminal domain termed AICD (tan). APP can be delivered to the plasma membrane through a canonical secretion pathway or delivered directly to early endosomes from the Golgi via an AP-4-dependent process. Cleavage of APP by α-Secretase on the plasma membrane constitutes a non-amyloidogenic processing pathway. APP can also be endocytosed into EEA1 and RAB5-positive vesicles that contain BACE1 and γ-Secretase. These organelles can mature to lysosomes that contain TMEM192 in the membrane and have been reported to contain higher levels of PSEN2[26], a catalytic component of γ-Secretase. The extent to which processing to form Aβ peptides occurs in the endosome or lysosome is unclear. **b** Schematic depicting the rational and workflow for detection of half-tryptic

peptides for quantification of peptides derived from Aβ40, 42, or 43. The sequence around the APP TM is shown, as well as the location of Aβ trigger peptides used for TOMAHAQ. Synthetic Aβ40, 42, and 43 peptides were digested with trypsin, labeled with TMT-126, mixed with TMT-unlabeled extracts from 293 cells, as well as with trigger peptides previously labeled with TMTsh. **c** During TOMAHAQ, trigger peptides identified in MS[2] are used to isolate "target" peptides from synthetic Aβ, which are then subjected to SPS-MS[3] to allow reporter ion quantification. MS[2] fragments used as trigger for the Aβ40 half-tryptic peptide are shown. **d** Signal-to-noise values for Aβ-derived tryptic and half-tryptic peptides derived from the experiment outlined in panel B (*n* = 1). **e** Summary of APP/Aβ trigger peptides, including their locations within the APP protein. Half-tryptic peptides for processing by BACE1 and γ-secretase are shown in red. Several peptides represent isoforms or phospho-forms of APP. **f** Chromatographic profiles of trigger peptides from MS[1].

Secretase machinery in endosomes and lysosomes. We observed comparable levels of both PSEN1 and 2 in both early/sorting endosomes and lysosomes when normalized to LAMP2 abundance, as determined by immunoblotting of Endo- and Lyso-IPs (Supplementary Fig. 6a). In contrast, BACE1 is enriched in endosomes, but comparatively reduced in lysosomes (Supplementary Fig. 6a), consistent with BACE1 being trafficked to recycling endosomes or degraded upon further trafficking in the endolysosomal system, as proposed previously[53].

In order to directly quantify processed Aβ in the endolysosomal system, we performed a series of 11-plex TOMAHAQ-TMT experiments examining the effect of BACE1 inhibitor (BSI, Lanabecestat) and γ-Secretase inhibitor (GSI, Semagacestat) on the abundance of Aβ cleavage products in samples subjected to LMW filtration (Fig. 6a, b and Supplementary Data 10). To monitor the signal-to-noise ratio for each peptide, which varied in magnitude with the particular properties of the peptide, we employed 293[EL]-APP[−/−] cells as a negative control for reporter ions (Fig. 6b). 293[EL]-APP* or 293[EL]-APP[−/−] cells were either left untreated, or treated with BSI or GSI (15 h) and subjected to Endo- and Lyso-IP (Fig. 6b). Aβ-derived half-tryptic peptides were analyzed after LMW filtration of organelles while extracellular and cytoplasmic peptides, including half-tryptic BACE1 cleavage products, were analyzed directly in organelle IPs, given that APP[FL] and CTFβ are largely found in the retentate, as schematized in Fig. 6a. As expected, BSI reduced the levels of CTFβ and increased the levels of CTFα in the post-nuclear supernatant (PNS) fraction while GSI led to increased levels of CTFα and CTFβ, as revealed by immunoblotting (Supplementary Fig. 6b). This pattern was largely maintained in both the Endo- and Lyso-IP samples (Fig. 6c and Supplementary Fig. 6b), which also indicated that CTFα/β remains associated with the organelle. We initially examined cleavage at the major BACE1 site (L652 in the Sw mutant), which would produce tryptic half-peptides E16 and E17 at the expense of the parental tryptic peptide E15 (Fig. 6d, e). In direct analysis of Endo-IPs, we identified both parental E15 peptide as well as BACE1 cleavage products E16 and E17 in samples from control DMSO-treated cells (Fig. 6e). Importantly, BSI treatment resulted in increased abundance of parental tryptic peptide E15, consistent with inhibition of APP processing by BACE1 (Fig. 6e). Concomitantly, the abundance of the two half-tryptic peptide products derived from BACE1 cleavage (E16 and E17) was reduced to levels comparable to that seen in APP[−/−] (KO) cells (Fig. 6e). A similar pattern was observed in Lyso-IPs (Fig. 6e).

For γ-Secretase products, Endo-IP_LMW samples 293[EL]-APP* cells contained readily detectable Aβ37, 38, 39, 40, and 42 with Aβ39, 40, and 42 being most abundant based on selective ion monitoring (SIM) of parent ions in MS[1] (Fig. 6f, Supplementary Data 11). Importantly, the signal/noise for these peptides was reduced to levels comparable to endosomes from APP[−/−] cells upon treatment with GSI (Fig. 6f). In all cases, BSI resulted in a reduction in the abundance of Aβ half-tryptic

peptides, as expected, since juxtamembrane processing is thought to be a pre-requisite for cleavage by γ-Secretase (Fig. 6f). In this setting, residual γ-Secretase half-tryptic peptides observed in the Endo-IP could reflect prior cleavage by α-Secretase, which is also expected to be permissive for subsequent cleavage by γ-Secretase. The relative abundance of Aβ40 and Aβ42 based on SIM is consistent with the use of the T700N mutant in APP, which promotes cleavage to produce Aβ42 at the expense of shorter forms (Fig. 6f). Routinely, signal-to-noise values for the Aβ34 peptide did not pass a *p* value cut-off for significantly changing with BSI or GSI treatment when compared with cells lacking APP (Fig. 6f), and we are therefore unable to quantify this peptide. Overall, similar patterns of Aβ cleavage products were seen with Lyso-IPs, although the absolute abundance of peptides within the lysosome, based on SIM scans, was 3–9-fold lower than seen with endosomes (Fig. 6f). Importantly, enrichment of Aβ peptides within Endo- or Lyso-IPs was needed to robustly detect Aβ peptides, as LMW filtration of the PNS allowed detection of only Aβ38 but with S/N comparable to cells lacking APP (Supplementary Fig. 6c). Moreover, analysis of Aβ peptides in Endo- or Lyso-IPs or the PNS without LMW filtration by TOMAHAQ proteomics resulted in detection of only a subset of Aβ peptides with typically very low S/N (Supplementary Fig. 6d). By comparison, we routinely detected peptides from extracellular and C-terminal domains within both Endo- and Lyso-IPs, as well as the PNS (Supplementary Fig. 7a–d and Supplementary Data 10). The relative abundance of these peptides was not dramatically altered in response to BSI or GSI.

## γ-Secretase modulator alters Aβ cleavage specificity in endosomes and lysosomes

Among Aβ peptides, Aβ40 and Aβ42 are thought to have the highest propensity to form amyloids[54]. As such, previous studies have sought to identify small molecules that shift processing of Aβ from these longer forms to shorter, less amyloidogenic Aβ peptides (including Aβ37/38), through allosteric modulation of the catalytic subunit of γ-Secretase[55]. One such γ-Secretase modulator (GSM) is BPN-15606. Previous studies have shown that GSM can enhance the cleavage of AICD by γ-Secretase in vitro to produce Aβ38 in either in vitro reactions, whole-cell extracts, or cell culture supernatants, as assessed typically using ELISA detection[56]. However, the question of whether GSMs could alter the extent of Aβ accumulation in endosomes and lysosomes was unknown. To examine this question, cells were treated with GSI or GSM (15 h) prior to Endo- or Lyso-IP (Fig. 7a). Immunoblotting of purified organelles revealed that while GSI produced the expected increase in CTFα and CTFβ within both endosomes and lysosomes, GSM had little effect relative to these compartments from untreated cells (Fig. 7b). Endo- and Lyso-IPs were subjected to Aβ filtration and then examined by TOMAHAQ proteomics (Fig. 7a, c, d, Supplementary Data 12). In endosomes, the abundance of Aβ39, 40, and 42 was reduced by GSM to an extent similar to that seen with GSI, approximating the level of background signal found in APP[−/−] cells (Fig. 7c, d). In contrast, Aβ37 and Aβ38 levels

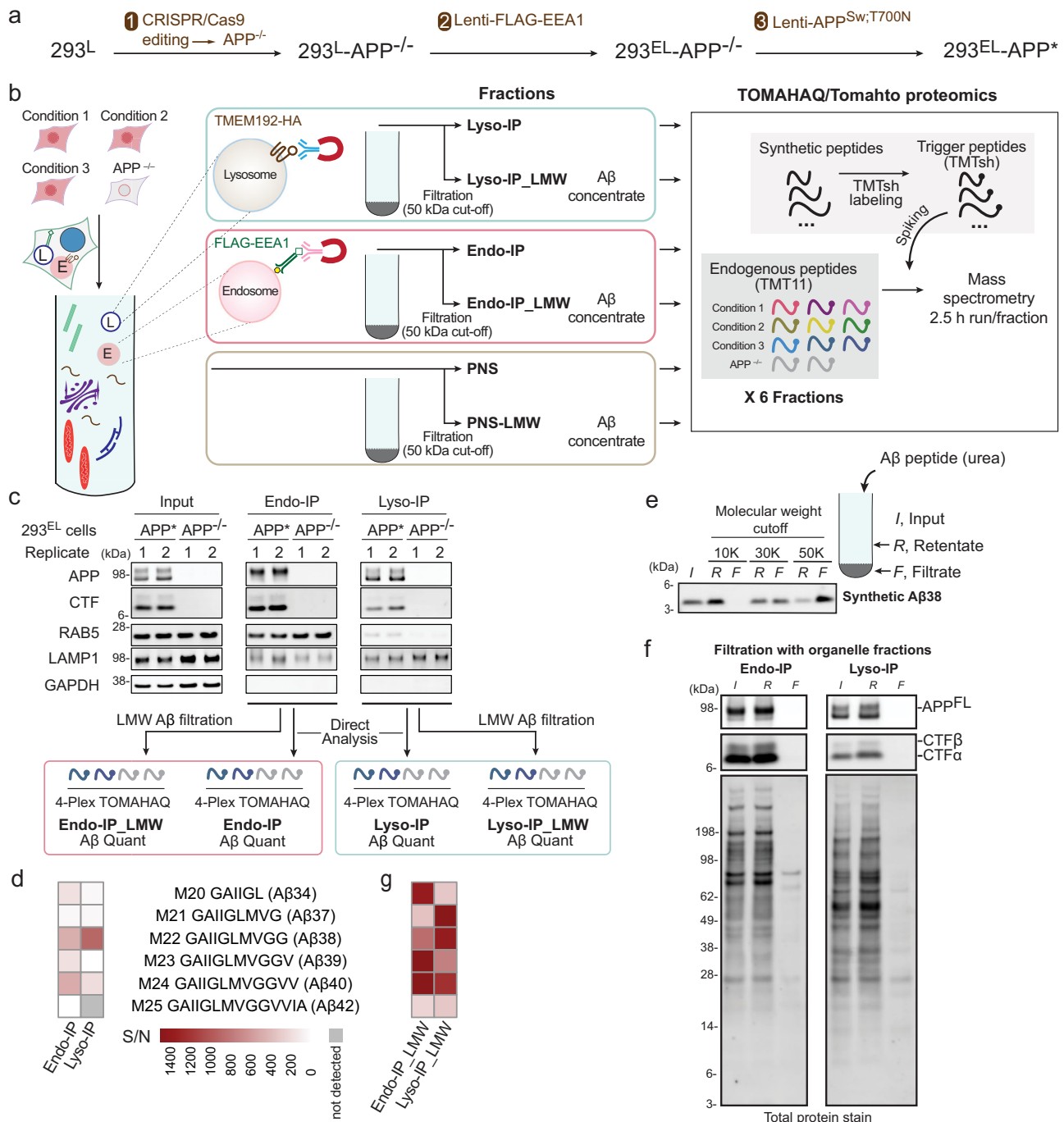

**Fig. 5 | Application of spatial endolysosomal proteomics to analysis of APP processing by γ-Secretase. a** *APP* in 293$^{EL}$ cells was deleted with CRISPR-Cas9 and APP$^{Sw;T700N}$ expressed stably using a lentivirus to create 293$^{EL}$-APP*. **b** The optimized workflow involves homogenization of cells to generate the post-nuclear supernatant (PNS) followed by Endo- or Lyso-IP, which are either used directly for TOMAHAQ proteomics via a multi-step workflow or solubilized in 8 M urea/0.5% NP-40 and LMW filtration using a 50 kDa filter to purify Aβ peptides for TOMAHAQ proteomics. **c** The indicated 293$^{EL}$ cells in biological duplicate (*n* = 2) were subjected to Endo- or Lyso-IP prior to immunoblotting (top panels) and analysis by TOMA-HAQ with or without LMW filtration. **d** TOMAHAQ-TMT reporter ion signal-to-noise ratios for samples from panel **c** (without filtration). Background intensities in APP$^{-/-}$

cells were subtracted from those of test samples. **e** Analysis of synthetic Aβ38 passage through filters with distinct molecular weight cut-offs (representative of 3 experiments). Synthetic Aβ in 8 M urea (with 0.5% NP-40) was applied to the filter and centrifuged for 12 min at 14,000 × *g*. Input, filtrate, and retentate samples were analyzed by SDS-PAGE and peptides stained with Total Protein Stain. **f** Proteins released from either Endo- or Lyso-IPs with 8 M urea/0.5% NP-40 were subjected to filtration with a 50 kDa cut-off filter and the input, retentate and filtrate analyzed by immunoblotting with the indicated antibodies or stain for total protein (representative of 3 experiments). **g** TOMAHAQ-TMT reporter ion signal-to-noise ratios for samples from panel **c** (with LMW filtration), as in panel **d**.

were increased ~2-fold compared with the steady-state abundance. Comparable results were found in an independent Endo-IP experiment (Supplementary Fig. 8a). The pattern of γ-Secretase cleavage products seen in Lyso-IPs paralleled that seen in Endo-IPs, albeit with lower

absolute levels of peptides as determined by SIM analysis (Supplementary Fig. 7d, Supplementary Data 13). Thus, Endo- and Lyso-IPs coupled with TOMAHAQ provide a means by which to spatially examine the specificity of Aβ processing by γ-Secretase.

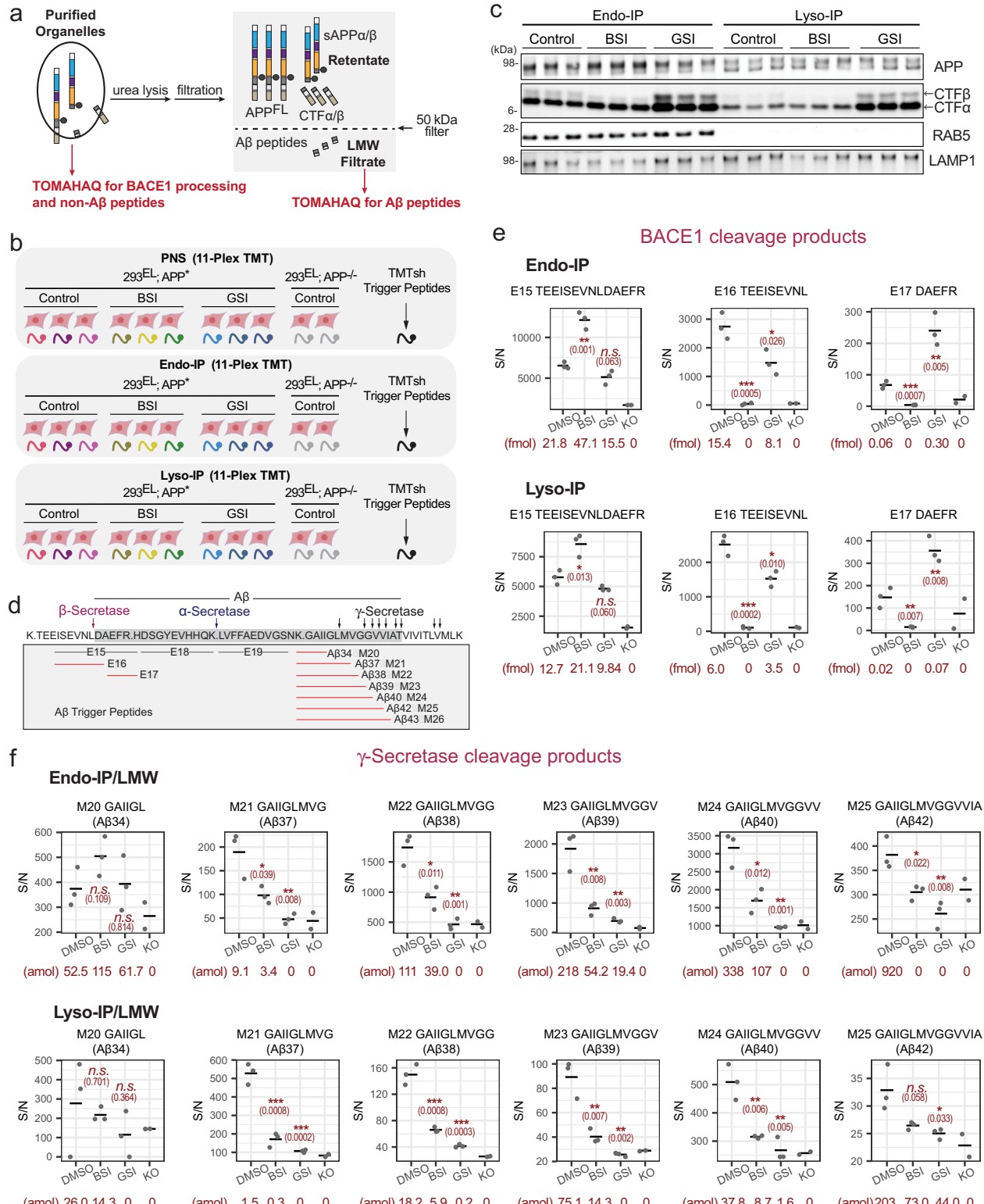

**a** Purified Organelles — urea lysis, filtration — Retentate: sAPPα/β, APP^FL, CTFα/β, Aβ peptides, 50 kDa filter, LMW Filtrate. TOMAHAQ for BACE1 processing and non-Aβ peptides. TOMAHAQ for Aβ peptides.

**c** Endo-IP / Lyso-IP — Control, BSI, GSI. APP, CTFβ, CTFα, RAB5, LAMP1.

**b** PNS (11-Plex TMT), Endo-IP (11-Plex TMT), Lyso-IP (11-Plex TMT). 293^EL; APP*, 293^EL; APP−/−. Control, BSI, GSI, Control. TMTsh Trigger Peptides.

**d** Aβ. β-Secretase, α-Secretase, γ-Secretase. K.TEEISEVNLDAEFR.HDSGYEVHHQK.LVFFAEDVGSNK.GAIIGLMVGGVVIAT.VIVITLVMLK. E15, E18, E19, E16, E17, Aβ34 M20, Aβ37 M21, Aβ38 M22, Aβ39 M23, Aβ40 M24, Aβ42 M25, Aβ43 M26. Aβ Trigger Peptides.

**e** BACE1 cleavage products

Endo-IP:
- E15 TEEISEVNLDAEFR — **(0.001), n.s.(0.063)** — (fmol) 21.8 47.1 15.5 0
- E16 TEEISEVNL — ***(0.0005), *(0.026) — (fmol) 15.4 0 8.1 0
- E17 DAEFR — ***(0.0007), **(0.005) — (fmol) 0.06 0 0.30 0

Lyso-IP:
- E15 TEEISEVNLDAEFR — *(0.013) — (fmol) 12.7 21.1 9.84 0
- E16 TEEISEVNL — ***(0.0002), *(0.010), n.s.(0.060) — (fmol) 6.0 0 3.5 0
- E17 DAEFR — **(0.007), **(0.008) — (fmol) 0.02 0 0.07 0

**f** γ-Secretase cleavage products

Endo-IP/LMW:
- M20 GAIIGL (Aβ34) — n.s.(0.109), n.s.(0.814) — (amol) 52.5 115 61.7 0
- M21 GAIIGLMVG (Aβ37) — *(0.039), **(0.008) — (amol) 9.1 3.4 0 0
- M22 GAIIGLMVGG (Aβ38) — *(0.011), **(0.001) — (amol) 111 39.0 0 0
- M23 GAIIGLMVGGV (Aβ39) — **(0.008), **(0.003) — (amol) 218 54.2 19.4 0
- M24 GAIIGLMVGGVV (Aβ40) — *(0.012), **(0.001) — (amol) 338 107 0 0
- M25 GAIIGLMVGGVVIA (Aβ42) — *(0.022), **(0.008) — (amol) 920 0 0 0

Lyso-IP/LMW:
- M20 GAIIGL (Aβ34) — n.s.(0.701), n.s.(0.364) — (amol) 26.0 14.3 0 0
- M21 GAIIGLMVG (Aβ37) — ***(0.0008), ***(0.0002) — (amol) 1.5 0.3 0 0
- M22 GAIIGLMVGG (Aβ38) — ***(0.0008), ***(0.0003) — (amol) 18.2 5.9 0.2 0
- M23 GAIIGLMVGGV (Aβ39) — **(0.007), **(0.002) — (amol) 75.1 14.3 0 0
- M24 GAIIGLMVGGVV (Aβ40) — **(0.006), **(0.005) — (amol) 37.8 8.7 1.6 0
- M25 GAIIGLMVGGVVIA (Aβ42) — n.s.(0.058), *(0.033) — (amol) 203 73.0 44.0 0

## Discussion

The ability to isolate intact lysosomes, mitochondria, peroxisomes, and synaptic vesicles through rapid and selective affinity enrichment has had a substantial impact on our understanding of these organelles, and especially their roles in metabolism and signaling[11,57–60]. The development of a rapid method for isolation of intact early/sorting endosomes, as described here, is expected to provide an alternative to gradient purification of EEA1-positive endolysosomal intermediates. EEA1 is recruited primarily to PI3P- and RAB5-positive endosomes, indicative of

early/sorting endosomes[3] (Fig. 4a). These organelles undergo protein sorting, fission to create RAB11-positive recycling endosomes, and fusion with other endosomes or trafficking vesicles as they mature into late endosomes, and ultimately, fully degradative lysosomes[3,61]. This maturation is accompanied by conversion of PI3P to PI(3,5)P$_2$ and loss of EEA1 binding (Fig. 4a). The ability of Endo-IP to facilitate access to this dynamic compartment is indicated by: (1) our ability to capture TF within isolated early endosomes as early as 5 min post TF addition, and (2) the collection of proteins that are enriched, including proteins

**Fig. 6 | Digital snapshots of APP/Aβ processing in early endosome and lysosomal compartments. a** Scheme highlighting distinct approaches for analysis of extracellular and cytosolic APP peptides versus analysis of Aβ peptides. Purified organelles are used for analysis of BACE1 processing and non-Aβ peptides while LMW filtrate is used for analysis of Aβ peptides. **b** Overview of experimental design. The indicated cells were left untreated or treated with BSI or GSI for 15 h followed by isolation of endosomes and lysosomes (see Fig. 5b). PNS samples were processed in parallel. Samples with or without LMW filtration were trypsinized and subjected to six sets of 11-plex TOMAHAQ analyses. **c** The indicated samples from panel **b** were subjected to immunoblotting with α-APP antibodies recognizing C-terminus of APP and CTFα,β. BSI and GSI treatments, representative of at least three independent experiments. **d** Schematic showing the sequences and peptides associated with APP cleavage by BACE1 and γ-Secretase. Half-tryptic Aβ trigger peptides are shown in red. **e** Quantitative analysis of BACE1 cleavage products in biological triplicate Endo-IP (upper panel) or Lyso-IP (lower panel) samples ($n = 3$). Signal-to-noise for MS$^3$ intensities are shown for each peptide. The signal associated

with samples from APP$^{-/-}$ cells ($n = 2$), considered as background, is shown. The center line represents average of the data points. Asterisks refer to two-sided Student's $t$-test of DMSO treated samples versus compound treatment: *n.s.*, not significant; *$p \le 0.05$; **$p \le 0.01$; ***$p \le 0.001$. Exact $p$ values are indicated in the parenthesis. Absolute abundance of individual peptides determined by SIM scans (see "Methods") is provided below each condition. **f** Quantitative analysis of γ-Secretase cleavage products in biological triplicate Endo-IP (upper panel) or Lyso-IP (lower panel) samples after LMW filtration ($n = 3$). Signal-to-noise for MS$^3$ intensities are shown for each peptide. The signal associated with samples from APP$^{-/-}$ cells, considered as background, is shown ($n = 2$). The center line represents average of the data points. Asterisks refer to two-sided Student's $t$-test of DMSO treated samples versus compound treatment: *n.s.*, not significant; *$p \le 0.05$; **$p \le 0.01$; ***$p \le 0.001$. Exact $p$ values are indicated in the parenthesis. Absolute abundance of individual peptides determined by SIM scans (see "Methods") is provided below each condition.

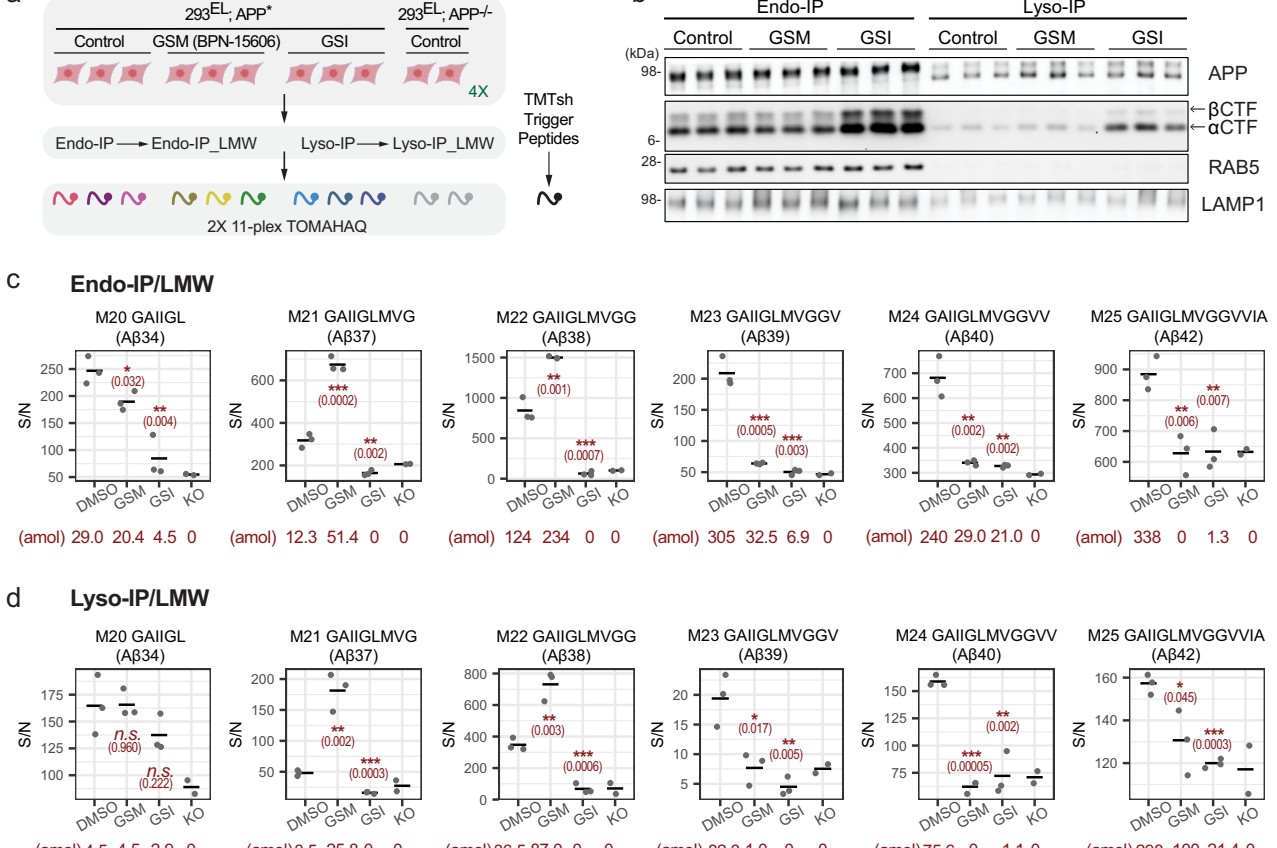

**Fig. 7 | Quantitative assessment of γ-secretase modulator action on APP in early endosomes and lysosomes. a** Overview of experimental design. The indicated cells ($n = 3$) were left untreated or treated with GSM (BPN-15606) and GSI for 15 h followed by isolation of endosomes and lysosomes together with LMW filtration. Samples were trypsinized and subjected to two sets of 11-plex TOMAHAQ proteomics. **b** The indicated biological triplicate samples from panel A were subjected to immunoblotting with the indicated antibodies. Quantitative analysis of Aβ peptides in biological triplicate Endo-IP (**c**) or Lyso-IP (**d**) samples after LMW

filtration ($n = 3$). Signal-to-noise for MS$^3$ intensities (relative to APP$^{-/-}$ cells, $n = 2$) is shown. Asterisks refer to two-sided Student's $t$-test of DMSO treated samples versus compound treatment: *n.s.* not significant; *$p \le 0.05$; **$p \le 0.01$; ***$p \le 0.001$. Exact $p$ values are indicated in the parenthesis. Absolute abundance of individual peptides determined by SIM scans (see "Methods") is provided below each condition. Absolute abundance of individual peptides determined by SIM scans (see "Methods") is provided below each condition.

involved in vesicle fusion and fission, protein sorting and maturation, and endocytic cargo. Endosomes are rich in sorting and vesicle fusion machinery, which are dramatically absent in lysosomes purified via Lyso-IP. In contrast, lysosomes are rich in lumenal enzymes that are largely absent from EEA1-positive endosomes. Endo-IP compared favorably with gradient purified endosomal fractions[32]; while similar numbers of endosomal proteins were identified by both methods,

components known to associate with late endosomes were present in gradient fractionated samples but were not enriched with Endo-IP (Fig. 2d, e). Recent studies have identified a small population of EEA1-positive late endosomes that lack detectable RAB5 and RAB7 but remain PI3P-positive[62]. Such a population could be potentially reflected in our ability to detect ESCRT-III on EEA1-associated vesicles. We note that a previous study[63] employed α-EEA1 for immunoprecipitation of

endosomes after gradient fractionation of organelles, but recovered less than 10% of the endosomal proteins isolated by our method, suggesting that direct and rapid immunoprecipitation may be more useful for some applications. Thus, the Endo-IP approach is complementary to Lyso-IP[11] and has the potential to facilitate a mechanistic understanding of distinct steps in the endolysosomal system, as exemplified here through analysis of APP processing.

While APP processing is known to occur within the endolysosomal system, multiple sometimes contradictory models have been put forward to describe where and how Aβ is formed from APP, particularly in the context of γ-Secretase[16,26,28]. We found that BACE1 and γ-Secretase cleavage products of APP can be detected within EEA1-positive endosomes, and the extent of cleavage at individual sites within Aβ is altered by γ-Secretase small-molecule modulators[56]. Moreover, γ-Secretase catalytic subunits PSEN1 and PSEN2 are present in EEA1-associated endosomes, and can presumably act together with BACE1 (also enriched in early/sorting endosomes) to process APP within this compartment. However, it is also plausible that partially processed APP could be delivered to EEA1-positive endosomes directly from the Golgi, given that under some conditions, BACE1 or γ-Secretase activity has been observed within the Golgi[16]. Although we detected Aβ products within the lysosome, further studies are required to understand the extent to which processing by Secretases continues in this compartment and how luminal Aβ accumulation may affect lysosomal function, as has been hypothesized[26,64].

Given that early endosomes are used for processing of numerous proteins captured from the plasma membrane, the use of Endo-IPs coupled with quantitative proteomics of half-tryptic peptides may provide a general route for elucidating the biochemical parameters for processing by endosomal juxta and intra-membrane proteases, including in neuronal cells.

## Methods
Detailed catalog information for individual reagents and materials can be found in Supplementary Data 14.

### Cell line construction and maintenance
Protocols for cell line construction can be found at: https://doi.org/10.17504/protocols.io.4r3l24kxxg1y/v2. HEK293 cells (from ATCC, CRL-1573), referred to throughout as "293", were maintained in Dulbecco' Modifies Eagles Medium (DMEM) with 10% fetal bovine serum and 1% Penicillin-Streptomycin. Additionally, 293EL cells were maintained in 1.2 µg/ml puromycin and 200 µg/ml G418. Gene editing was performed as described[65]. For endogenous tagging of TMEM192 with 3xHA, cells were co-transfected with pX459 containing a gRNA (5′-AGTAGAA CGTGAGAGGCTCA) targeting adjacent to the termination sequence in TMEM192 and pSMART containing 5′ and 3′ homology arms for TMEM192 in which the termination codon is replaced by a 3xHA epitope sequence followed by a TAA stop codon[31], except that the puromycin resistance cassette was replaced by a neomycin resistance cassette. Homozygously targeted clones were identified by immunoblotting cell extracts with α-HA and α-TMEM192[31]. These cells are referred to as 293L (RRID#: CVCL_C0I5).

Puromycin-resistant pHAGE lentiviral vectors expressing EEA1, RAB11A, TFR1, and RAB5A were generated by recombining open reading frames in pENTR vectors from ORFEOME8.1[66] into a pHAGE-N-3xFLAG vector. Viral supernatants derived from transfection of pHAGE vectors into 293 T cells were used to infect the indicated cell lines. Puromycin (1.2 µg/ml) was used to selected for viral integration. 293L cells expressing Flag-EEA1 are referred to as 293EL (RRID#: CVCL_C0I7).

For APP knock-out, oligonucleotides (Top: 5′-CACCGGTCAACGG CATCAGGGGTAC, Bottom: 5′-AAACGTACCCCTGATGCCGTTGACC) were phosphorylated, annealed, and cloned into a pX459 vector[65]. 293L cells were transfected with the plasmid by Lipofectamine 3000 and selected with 1.2 µg/mL of puromycin. Monoclonal cells were grown,

and deletion of the APP was confirmed by immunoblotting. 293L cells lacking APP are referred to as 293L-APP−/− (RRID#: CVCL_C0I6). 293EL cells lacking APP are referred to as 293EL-APP−/− (RRID#: CVCL_C0I8).

To create an APP (isoform 751) open reading frame, pENTR-APP751 (open, no stop codon; http://dnasu.org/DNASU/GetCloneDetail.do?cloneid=351686), was amplified by PCR to replace W752 with a stop codon using forward primer (5′-GCAGAAC**TAG**ATC CACCCAGCTTTCTTG) and reverse primer (5′-GGGTGGAT**CTA**GTTCT GCATCTGCTCAAAG). pENTR-APP^Sw,T700N was generated by two rounds of PCRs using the following kits and primers: Sw (K651N/N652L), QuickChange II mutagenesis kit, Forward: 5′-TCGGAATTCTGCATCC**CA GATT**CACTTCAGAGATCTCCTCCG, Reverse: 5′-CGGAGGAGATCTCT GAAGTG**AATCT**GGATGCAGAATTCCGA; T700N, Q5 mutagenesis kit, Forward: 5′-ATCGTCATC**AAC**TTGGTGATG, Reverse: 5′-CACTGTCGC TATGACAAC. The APP^Sw,T700N open reading frame in pENTR was transferred to Gateway destination vector pHAGE-C-HA-FLAG-puro using LR Clonase to yield pHAGE-APP^Sw,T700N -puro. Note: the stop codon in the APP open reading frame blocks translation into the HA-FLAG tag in this vector. Stably expressing APP^Sw,T700N cell line (referred to as 293EL-APP*; RRID#: CVCL_C0I9) was prepared by lentiviral transduction to 293EL-APP−/− followed by monoclonal selection.

### Western blotting
A protocol for immunoblotting methods used here can be found at: https://doi.org/10.17504/protocols.io.kqdg36jeeg25/v2. Briefly, samples were lysed either by homogenization in KPBS buffer, urea buffer, or RIPA buffer with protease and phosphatase inhibitors. Total protein concentration was determined by BCA or Bradford assay, samples were normalized with additional buffer, and samples were combined with NuPAGE LDS buffer (4×) plus NuPAGE reducing agent (10×). Samples were loaded onto 4-12% NuPAGE Bis-Tris gels and separated by electrophoresis in MES buffer. Proteins were transferred to PVDF or nitrocellulose membranes by standard wet transfer in 20% methanol. Membranes were stained with REVERT 700 total protein stain following manufacturer's instructions, and total protein was imaged with a ChemiDoc MP at 680 nm. After de-staining with REVERT reversal solution for 5 min, membranes were blocked with tris-buffered saline (TBS) (5% non-fat dry milk) at room temperature for 60 min. Membranes were incubated overnight at 4 °C with primary antibody solution in TBS with 0.1% Tween-20 (TBST), washed six times with TBST for 5 min each, and incubated in secondary antibody solution in TBST (plus 0.01% SDS) for 1 h at room temperature. Membranes were washed four times with TBST for 5 min each. When using HRP-conjugated secondary antibodies, luminol and hydrogen peroxide solution was applied to membrane for 2 min, and membrane were imaged with a ChemiDoc MP using the chemiluminescent setting. When using Li-Cor fluorescent secondary antibodies, membranes were blotted dry and imaged with a ChemiDoc MP at either 800 nm or 680 nm, depending on the secondary antibody. Data were analyzed with ImageLab v6.0.1.

### Immunofluorescence
Colocalization of RAB5 and FLAG-EEA1 was assessed by standard immunofluorescence, as described in the following protocol: https://doi.org/10.17504/protocols.io.ewov146xkvr2/v2. Briefly, No.1.5 coverslips were coated in 0.01% poly-ʟ-lysine solution and incubated for 15 min at 37 °C. Poly-ʟ-lysine was aspirated, and coverslips were washed twice with sterile water and dried for 15 min at 37 °C. 293 or 293EL cells were seeded into several wells each to be approximately 70% confluent the next day. For the dynamin inhibition experiment (relevant to Supplementary Fig. 2f), cells were treated with either DMSO (0.4%) or Hydroxy Dynasore (Dyngo4a) (20 µM final) in serum-free DMEM for 3 h. After treatment, cells were washed with DMEM with 10% serum and 0.4% DMSO and then with DPBS. Cells were fixed in 4% paraformaldehyde in DPBS for 15 min at 25 °C. Samples were washed three

times with DPBS and blocked with blocking solution (1% BSA, 0.15% Triton X-100 in DPBS) for 1 h at 25 °C. Blocking solution was removed, and samples were incubated in primary antibody solution (α-RAB5 at 1:200 and α-DYKDDDDK at 1:200 in blocking solution) overnight at 4 °C. Samples were washed three times with blocking solution, then incubated in secondary antibody solution (Goat α-Rabbit-594 at 1:400 and Goat α-Mouse-488 at 1:400) for 1 h at 25 °C and protected from light. Samples were stained with 1.25 μg/mL Hoechst 33342 solution in DPBS for 10 min at 25 °C and protected from light. Samples were washed three times with blocking solution, washed once with DPBS, mounted on slides with ProLong Glass Antifade Mountant. Cells were imaged using a Yokagawa CSU-X1 spinning disk confocal on an inverted Nikon Ti fluorescence microscope with a Nikon Plan Apo 100×/1.45 NA oil objective lens. Under control by an AOTF (Spectral Applied Research LMM-6 laser merge module), Alexa Fluor 488 and 594 fluorophores were excited by 488 nm (100 mW) and 561 nm (100 mW) solid state lasers, respectively. The emission was collected with Semrock Di01-T405/488/568/647 dichroic mirror and Chroma ET525/50-nm or ET620/60-nm emission filters. Wide-field fluorescence images of Hoechst were collected using Lumecon SOLA fluorescence light source with Chroma 395/25x excitation filter, 400dclp dichroic mirror, and ET460/50-nm emission filters. Confocal and wide-field images were acquired by Hamamatsu ORCA-ER cooled CCD camera and ORCA-R2 cooled CCD camera, respectively, controlled with Meta-Morph v7.10 image acquisition software. Fiji was used to adjust brightness, contrast, and gamma and to analyze the images[67]. Mander's correlation coefficients were calculated with JACoP plugin to assess the colocalization of signals from two channels. For analysis of TF uptake, cells were serum starved for 1 h and then treated with TF-Alexa-647 (25 μg/mL) in serum-free media. Cells were washed and fixed at the indicated times prior to immunostaining with α-FLAG antibodies and imaged by light microscopy as described above.

## Organelle immunoprecipitation

**Lysosomal immunoprecipitation (Lyso-IP) for organelle proteomics.** Lyso-IPs were performed as described[31] and as detailed in https://doi.org/10.17504/protocols.io.ewov14pjyvr2/v2. 293 or 293[EL] cells were seeded in 15-cm dishes, with one dish per replicate. At 80% confluency the cells were harvested on ice by scraping in 2 mL of DPBS and pelleting at $1000 \times g$ for 2 min at 4 °C. The supernatants were discarded, and the pellets were washed once with 1 mL of cold KPBS buffer (25 mM KCl, 100 mM potassium phosphate, pH 7.2) and pelleted at $1000 \times g$ for 2 min at 4 °C. Cell pellets were resuspended in 1 mL of KBPS buffer supplemented with protease and phosphatase inhibitor tablets and lysed with 30 strokes with a 2-mL Dounce homogenizer on ice. The lysed cells were centrifuged at $1000 \times g$ for 5 min at 4 °C, and the post-nuclear supernatants (PNS) were transferred to new tubes on ice. The protein concentration of each lysate was determined by Bradford assay, and 10 μL of each PNS was transferred to a new tube and combined with 20 μL of RIPA lysis buffer and 10 μL of 4× LDS buffer with reducing agent for later analysis by Western blot. α-HA magnetic beads (60 μL of bead slurry per dish) were washed three times with 1 mL KPBS buffer and resuspended in the same buffer. The resuspended bead slurry was added to each PNS, and samples were incubated at 4 °C for 50 min with gentle rotation. The beads were separated from the lysate with a magnetic stand, and the flow through was collected. For Western blot analysis, 10 μL of each flow through was combined with 20 μL of RIPA lysis buffer and 10 μL of 4× LDS buffer with reducing agent. Using a magnetic stand, the beads were washed twice with 500 μL of high-salt KPBS buffer (25 mM KCl, 100 mM potassium phosphate, 155 mM NaCl, pH 7.2) with protease and phosphatase inhibitors cocktail, then washed once with KPBS with inhibitors. Samples were eluted by addition of 120 μL 0.5% NP-40 in KBPS with inhibitors for 30 min at 4 °C with gentle rotation. For Western blot analysis, 20 μL of each eluate was combined with 6.7 μL of 4× LDS

buffer with reducing agent. The remainder of the eluates were either immediately processed or snap frozen in liquid nitrogen and stored at −80 °C until processing for mass spectrometry.

**Endosomal immunoprecipitation (Endo-IP) for organelle proteomics, transmission electron microscopy, and lipidomics.** Endo-IPs were performed as described (https://doi.org/10.17504/protocols.io.ewov14pjyvr2/v2). Endo-IPs were performed essentially as described for Lyso-IP, using 293[EL] cells expressing FLAG-EEA1 and control 293 cells, as detailed in https://doi.org/10.17504/protocols.io.byi9puh6. Briefly, 293[EL] and 293 cells were seeded in 15-cm dishes with one dish per replicate. If treating with DNM1/2 inhibitor Hydroxy Dynasore, 70–80% confluent dishes were treated with either DMSO (0.4%) or Hydroxy Dynasore (20 μM final) in serum-free DMEM for 3 h. Cells were harvested at 70–80% confluency on ice by scraping in 2.5 mL DPBS and pelleting at $1000 \times g$ for 2 min at 4 °C. The supernatants were discarded, and the pellets were washed once with 1 mL of KPBS buffer (25 mM KCL, 100 mM potassium phosphate, pH 7.2) and pelleted at $1000 \times g$ for 2 min at 4 °C. Cell pellets were resuspended in 500 μL of KPBS supplemented with protease inhibitor cocktail and PhosSTOP tablets and lysed with 30 strokes with a 2-mL Dounce homogenizer on ice. The lysed cells were centrifuged at $1000 \times g$ for 5 min at 4 °C, and the post-nuclear supernatants (PNS) were transferred to new tubes on ice. The protein concentration of each lysate was determined by Bradford assay, and 10 μL of each PNS was transferred to a new tube and combined with 20 μL of RIPA lysis buffer and 10 μL of 4× LDS buffer with reducing agent for later analysis by Western blot. α-FLAG M2 magnetic beads (60 μL of bead slurry per dish) were washed three times with 1 mL KPBS buffer. The resuspended bead slurry was added to each PNS, and samples were incubated at 4 °C for 50 min with gentle rotation. The beads were separated from the lysate with a magnetic stand, and the flow through was collected. For western blot analysis, 10 μL of each flow through was combined with 20 μL of RIPA lysis buffer and 10 μL of 4× LDS buffer with reducing agent. Using a magnetic stand, the beads were washed twice with 500 μL of KPBS (25 mM KCl, 100 mM potassium phosphate, pH 7.2) with protease and phosphatase inhibitors cocktail, then washed once with KPBS with inhibitors. Under these standard immunoprecipitation conditions, we recover ~2.5% of the EEA1 present in total cell extracts, which is similar to the 3% recovery of lysosomes as reported previously[11,57]. For analysis by negative stain transmission electron microscopy, samples were eluted by addition of 50 μL FLAG peptide solution (500 μg/mL in KBPS) at 25 °C for 45 min with gentle shaking. Eluates were transferred to new tubes, and 25 μL of each eluate was submitted to the Harvard Medical School Electron Microscope Facility. Alternatively, for analysis by mass spectrometry, samples were eluted by addition of 120 μL 0.5% NP-40 in KBPS with inhibitors for 30 min at 4 °C with gentle rotation. For Western blot analysis, 20 μL of each eluate was combined with 6.7 μL of 4× LDS buffer with reducing agent. The remainder of the eluates were either immediately processed or snap frozen in liquid nitrogen and stored at −80 °C until processing for LC−MS. Alternatively, samples can be processed for lipidomics, as described below. In some experiments, serum was withdrawn from cells for 1 h prior to re-feeding with serum containing 25 μg/mL of TF. At the indicated times, cells were scraped with cold PBS on ice, then washed with lysis buffer for Endo-IP, followed by either analysis by proteomics or by immunoblotting.

**Endolysosomal preparation for APP/Aβ TOMAHAQ proteomics.** Endo-IP and Lyso-IPs for TOMAHAQ proteomics were performed as described (https://doi.org/10.17504/protocols.io.ewov14pjyvr2/v2). For each replicate, 293[EL]-APP* cells were seeded in 5 × 15 cm dishes (2 × 15 cm for Lyso-IP and 3 × 15 cm for Endo-IP), and 293[EL]-APP[−/−] cells were seeded in 5 × 15 cm dishes so that they were approximately 60% confluent the next day and approximately 80–90% confluent two days later. Generally, three replicates of each 293[EL]-APP* treatment group

(e.g. DMSO or secretase inhibitors) and two replicates of 293$^{EL}$-APP$^{-/-}$ were processed simultaneously. One day after seeding, cells were treated with vehicle control (DMSO), GSI, GSM, or BSI to a final concentration of 2 μM and 0.2% DMSO. Cells were incubated with the compounds for 15 h. The next day, cells were harvested by discarding media and scraping in 2 mL KPBS buffer supplemented with DMSO, GSI, GSM, or BSI (note that the appropriate compound was used in KPBS buffer throughout subsequent steps to continue inhibiting the desired enzyme). Cells were pelleted at $1000 \times g$ for 2 min at 4 °C, supernatants were discarded, pellets were gently resuspended in 5 mL KPBS, and cells were pelleted at $1000 \times g$ for 2 min at 4 °C. Pellets were resuspended in 5 mL of KPBS with the addition of protease and phosphatase inhibitors and lysed with 20 strokes with a 7-mL Dounce homogenizer and tight pestle. The lysate was clarified by centrifugation at $1000 \times g$ for 5 min at 4 °C. The lysate may be further clarified by transferring the PNS from the first spin to a new tube on ice, spinning again, and transferring the final PNS to a new tube. The protein concentration of each lysate was determined by Bradford assay, and 10 μL of each PNS was transferred to a new tube and combined with 20 μL of RIPA lysis buffer and 10 μL of 4× LDS buffer with reducing agent for later analysis by Western blot. One hundred and ten microliters of each PNS was combined with 183 μL of 8 M urea/50 mM NaCl/0.8% NP-40 buffer and stored at −80 °C for later analysis by mass spectrometry.

α-FLAG M2 and α-HA magnetic beads (50 μL of bead slurry per dish) were prepared on a magnetic stand by washing three times with KPBS and resuspending in KPBS (25 μL per dish for α-FLAG M2 beads and 50 μL per dish for α-HA beads). One hundred and fifty microliters of α-FLAG M2 beads were added per PNS (which came from $3 \times 15$ cm dishes) and 100 μL of α-HA beads were added per PNS (which came from $2 \times 15$ cm dishes). Samples were incubated for 45 min at 4 °C with gentle rotation. The beads were separated from the flow through with a magnetic stand, and the flow through was collected. For Western blot analysis, 10 μL of each flow through was combined with 20 μL of RIPA lysis buffer and 10 μL of 4× LDS buffer with reducing agent. α-FLAG beads were washed twice with 500 μL KPBS (with the compound) and then once with 1 mL KPBS (without compounds). α-HA beads were washed twice with 500 μL high-salt KPBS (KPBS with 155 mM NaCl and the compound) and once with KPBS (without compounds). Samples were eluted with 5 M urea/0.5% NP-40 KPBS buffer (180 μL for α-FLAG beads and 120 μL for α-HA beads) for 50 min at 30 °C with shaking. For Western blot analysis, 10 μL of each eluate was combined with 3.3 μL of 4× LDS buffer with reducing agent. The remainder of each eluate was split in two for future "Lyso" or "Endo" (20% of eluate) and "Lyso_LMW" or "Endo_LMW" (80% of eluate) samples, the latter of which were filtered through Amicon Ultra 0.5 mL 50 kDa centrifugal filters as follows to detect low abundance Aβ peptides. 250 μL of each PNS was loaded onto the Amicon column, and the remainder of the PNS was reserved to serve as the regular PNS sample. Lyso_LMW samples were diluted with 112 μL of 5 M urea/0.5% NP-40 buffer and loaded onto the columns. Endo_LMW samples were diluted with 64 μL of 5 M urea/0.5% NP-40 buffer and loaded onto the columns. Columns were centrifuged at 14,000×g at 10 °C for 12 min or until residual column volume was approximately 50 μL. To increase the yield of filtered Aβ peptides, the residual retentate was diluted with 150 μL of 5 M urea/0.5% NP-40 buffer, and the column was centrifuged at $14,000 \times g$ at 10 °C for 12 min. The final filtrate volume was measured and transferred to new Protein LoBind tubes. Remaining, unfiltered PNS, Lyso, and Endo samples were each diluted with 20 μL 5 M urea/0.5% NP-40.

All samples (unfiltered PNS, Lyso, and Endo & Amicon-filtered PNS_LMW, Lyso_LMW, and Endo_LMW) were reduced by addition of TCEP to 5 mM final and incubated at 25 °C for 30 min. Cysteines were alkylated by addition of iodoacetamide to 15 mM final and incubated at 25 °C for 30 min and protected from light. Samples were diluted with 50 mM EPPS buffer for 1.2 M urea final concentration. Proteins were precipitated by addition of 6.1 N TCA solution to 20% final and incubation at 4 °C for 1.5 h. Samples were centrifuged at $21,000 \times g$ for 15 min at 4 °C, and supernatants were removed. Samples were washed twice with ice-cold acetone by centrifuging at $21,000 \times g$ for 10 min at 4 °C. After final wash, the protein pellets were briefly dried in a SpeedVac. Pellets were resuspended in 10 μL of 8 M urea buffer followed by sonication in a water bath sonicator, and urea was diluted by addition of 10 μL of 200 mM EPPS. Peptides were digested by addition of 0.3 μg of LysC and incubated at 37 °C for 2 h. Urea was further diluted to 1.6 M final by addition of 200 mM EPPS. Peptides were digested by addition of 0.4 μg trypsin and incubated at 37 °C overnight. The next day, acetonitrile (ACN) was added to 30% final, and the peptides were reacted with 3.6–5.3 μL of the TMT 11-plex reagents (10 μg/μL in anhydrous ACN) for 1 h at 25 °C. Labeling was quenched by addition of hydroxylamine to 0.5% final followed by incubation at room temperature for 15 min. Pooled sample was dried by SpeedVac and desalted by C18 StageTip. The samples were resuspended with 5% ACN/8% formic acid (FA). Synthetic reference peptides labeled with TMTsh (super-heavy) were added to sample and analyzed by TOMA-HAQ as described below.

## Synthetic peptides

APP peptides corresponding to extracellular and cytosolic regions were designed based on data available in Peptide Atlas suggesting favorable LC−MS properties. Half-tryptic peptides were designed based on the desired cleavage sites for BACE1 and γ-Secretase. The sequences and properties of all APP peptides are provided in Supplementary Data 7 and were synthesized commercially by Biomatik and Thermo Fischer Scientific. Vendor-provided quality control for peptides indicated that all peptides were greater than 95% purity based on HPLC and had the expected mass based on MALDI-TOF while M25 (GAIIGLMVGGVVIA) and M26 (GAIIGLMVGGVVIAT) peptides had 88% and 74% purity, respectively. Peptides were reconstituted with 3% ACN/0.1% FA, and the concentration was quantified using Pierce Quantitative Fluorometric Peptide Assay. Ten microliters of 200 μM each peptide in 200 mM EPPS buffer (pH 8.5) was mixed with 1 μL of 5 μg/μL super-heavy TMT reagent (TMTsh), and the labeling efficiency was confirmed by LC−MS. M25 and M26 peptides were labeled with TMTsh in 50% EPPS/50% DMSO solvent. Extra TMTsh was added to under-labeled peptides to reach > 95 % labeling on both N-termini and lysine residues. After being desalted with C18 StatgeTip, the labeled peptides were reconstituted in 5% ACN/8% FA, and the reconstituted peptides were pooled. A protocol for handling of synthetic APP peptides can be found at https://doi.org/10.17504/protocols.io.bp2l6bqk1gqe/v2.

## Lipidomics

Lipidomics was performed as described in https://doi.org/10.17504/protocols.io.byn2pvge. Briefly whole-cell pellet or endosome-bound beads were thawed on ice. Once thawed, 60 μL of methanol, 200 μL of methyl-*tert*-butyl ether (MTBE), and 50 μL of water were added. Samples were vortexed for 10 s. The samples were then sonicated for 5 min using a program of 20 s on, 10 s off, and an amplitude of 30 (Qsonica, chilled bath sonicator). The temperature was maintained at 14 °C during sonication. After centrifugation for 10 min at 10,000 g at 4 °C, 150 μL, or 100 μL for endosome vs. lysosome comparison, of the lipophilic (upper) layer from the biphasic extraction was aliquoted into a separate glass vial, dried down by vacuum concentrator for 90 min. Samples were resuspended in 50 μL of resuspension solvent, either acetonitrile/isopropyl alcohol/water (ACN/IPA/water, 65:30:5, v/v/v) when comparing endosomes *vs.* whole cells or methanol/toluene, 9:1 v/v when comparing endosomes *vs.* lysosomes, and vortexed.

Sample analysis was performed by LC−MS, lipids were separated on an Acquity CSH C18 column held at 50 °C (100 mm × 2.1 mm × 1.7 μm particle size; Waters) using a Vanquish Binary Pump (400 μL/min

flow rate; Thermo Scientific). Mobile phase A was 10 mM ammonium acetate in ACN:H$_2$O (70:30, v/v) containing 250 μL/L acetic acid, and Mobile phase B was 10 mM ammonium acetate in IPA:ACN (90:10, v/v) with 10 mM ammonium acetate and 250 μL/L acetic acid. Initially, Mobile phase B was initially held at 2% for 2 min, then increased to 30% over 3 min, then increased to 50% over 1 min, then to 85% over 14 min, and finally to 99% over 1 min and held at 99% for 7 min. Mobile phase B was returned to 2% for 1.75 min before the next injection. 10 μL of extract was injected by a Vanquish Split Sampler HT autosampler (Thermo Scientific).

The LC system was coupled to a Q Exactive Orbitrap HF mass spectrometer through a heated electrospray ionization (HESI II) source (Thermo Scientific). Source conditions were as follow: HESI II and capillary temperature at 350 °C, sheath gas flow rate at 25 units, aux gas flow rate at 15 units, sweep gas flow rate at 5 units, spray voltage at | 3.5 kV| for both positive and negative modes. S-lens RF was set at 90.0 units for endosome vs. whole-cell comparisons and set to 60.0 units for endosome vs. lysosome comparisons. The MS was operated in a polarity switching mode acquiring positive and negative full MS and MS$^2$ spectra (Top2) within the same injection. Acquisition parameters for full MS scans in both modes were 30,000 resolution, $1 \times 10^6$ automatic gain control (AGC) target, 100 ms ion accumulation time (max IT), and 200–1600 $m/z$ scan range (or 200–2000 $m/z$ scan range for endosome vs. lysosome comparison). MS$^2$ scans in both modes were then performed at 30,000 resolution, $1 \times 10^5$ AGC target, 50 ms max IT, 1.0 $m/z$ isolation window, stepped normalized collision energy (NCE) at 20, 30, 40, and a 10.0 s dynamic exclusion.

The resulting LC–MS data were processed using Compound Discoverer 2.1 or 3.1 (Thermo Scientific) and Lipidex v1.1 platforms[42,43]. All peaks with a 0.4 min to 21 min retention time and 100 Da to 5000 Da MS$^1$ precursor mass were aggregated into distinct chromatographic profiles (i.e., compound groups) using a 10-ppm mass and 0.4 or 0.5 min retention time tolerance. Profiles not reaching a minimum peak intensity of $5 \times 10^5$, a maximum peak width of 0.75, a signal-to-noise (S/N) ratio of 1.5, and a fivefold intensity increase over blanks were excluded from further processing. MS/MS spectra were searched against in-silico generated lipid spectral librarys "Lipidex_HCD_acetate" and an in-house build BMP library with a MS1 search tolerance of 0.005 $m/z$. Spectral matches with a dot product score greater than 500 and a reverse dot product score greater than 700 were retained for further analysis. Phosphoglycerol lipid annotations were only retained if dot product score and reverse dot product score was greater than 700. Lipid MS/MS spectra that contained <75 % interference from co-eluting isobaric lipids, eluted within a 3.5 median absolute retention time deviation (M.A.D. RT) of each other, and found within at least 2 processed files were then identified at the molecular species levels, otherwise lipids are reported at species level (see also LIPID MAPS nomenclature[68]). Finally, identifications were manually evaluated and further filtering was preformed if lipid identifications were not consistent with expected effective carbon number vs. retention time models.

Lipid relative quantification values are integrated peak area of a MS1 peak. The particular quantification ion is selected as the ion which is most consistently observed across all samples. Using this approach, only one ion for a specific lipid feature would be represented in Supplementary Data 6 (either positive or negative, but not both) as additional adducts, isotopes, or in source fragments are also detected and excluded from quantification, see also LipiDex manuscript[50]. Although a single ion is used for relative quantification of a specific lipid, the lipid identifications leverage all suitable evidence about an MS1 feature – including MS/MS in positive and negative. All annotated spectra used for lipid identification can be found in Supplementary Data 1; these files were generated using code available on GitHub (https://github.com/coongroup/LipiDexSpectrumAnnotator; https://doi.org/10.5281/zenodo.7086361).

Lipid relative quantitation values (integrated chromatographic peak areas) were exported and analyzed by R 3.6.3. Two-sided Student's $t$-test was performed by $t\_test()$ function in the $rstatix$ package version 0.7.0. Individual $p$ values were adjusted for multiple testing correction, which was done by $mt.rawp2adjp()$ function with two-stage Benjamini & Hochberg (2006)[69] step-up FDR-controlling procedure in the $multtest$ package version 2.42.0.

## Proteomics

**Whole-cell global proteomics.** Whole-cell proteomics of 293 and 293$^{EL}$ cells was performed essentially as described (https://doi.org/10.17504/protocols.io.bys6pwhe). Each cell line was seeded in one 15-cm dish then harvested by scraping. After being washed with DPBS, cell pellets were resuspended with 8 M urea buffer supplemented with protease and phosphatase inhibitors and lysed by sonication. The lysates were centrifuged at 17,000 × $g$ for 8 min at 4 °C, and the supernatant was collected. Total protein concentration was estimated using a BCA assay, and 50 μg of the sample was reduced by addition of TCEP to 5 mM final and incubated at 25 °C for 30 min. Cysteines were alkylated by addition of iodoacetamide to 15 mM final and incubated at 25 °C for 30 min protected from light. After alkylation, the sample was diluted in EPPS buffer for 1 M urea final concentration. Proteins were precipitated by addition of 6.1 N TCA solution to 20% final and incubated at 4 °C for 1 h. Samples were centrifuged at 20,000 × $g$ for 7 min at 4 °C, and supernatants were removed. Pellets were washed twice with ice-cold acetone by centrifuging at 20,000 × $g$ for 10 min at 4 °C for 1 h. After final wash, the protein pellets were briefly dried in a SpeedVac. Pellets were resuspended in 50 μL of 8 M urea buffer, followed by sonication in a water bath sonicator, and urea was diluted by addition of 50 μL of 200 mM EPPS, pH 8.5. Peptides were digested by addition of 1 μg of LysC and incubated at 30 °C for 4 h. Urea was further diluted to 1.6 M final by addition of 200 mM EPPS. Peptides were further digested by addition of 1 μg trypsin and incubated at 37 °C overnight. The next day, ACN was added to 30% final, and the peptides were reacted with 10 μL of the TMTpro reagents (12.5 μg/μL in anhydrous ACN) for 1 h at 25 °C. Labeling was quenched by addition of hydroxylamine to 0.5% final followed by incubation at room temperature for 15 min. TMT-labeled samples were pooled with 1:1 ratio, and the mixture was dried and subjected to C18 solid-phase extraction using Sep-Pak.

The sample was resuspended in 110 μL of 10 mM ammonium bicarbonate/5% ACN and filtered through a 0.2 μm PTFE centrifugal filter. The sample was pre-fractionated by high-pH reverse-phase HPLC (Agilent 1260 Infinity) with an Aeris C18 column (250 mm × 4.6 mm) with a gradient of mobile phase A (10 mM ammonium bicarbonate, 5% ACN) and mobile phase B (10 mM ammonium bicarbonate, 90% ACN). 96 fractions were collected between 10 min (10% mobile phase B) and 72 min (100% mobile phase B) at a flow rate of 0.6 mL/min, and the fractions were concatenated into 24 fractions. Fractions were dried by SpeedVac and desalted by C18 StageTip. Alternating 12 fractions out of the 24 fractions were resuspended with 3% ACN/1% FA for mass spectrometry.

Mass spectrometry was performed by an Orbitrap Eclipse coupled with a Proxeon EASY-nLC1200 liquid chromatography pump. Peptides were separated on a microcapillary column (100 μm inner diameter) packed with ~35 cm of Accucore150 resin (2.6 μm, 150 Å, Thermo Fisher Scientific) with 8–23% (3–73 min), 23-30% (73–80 min), 30–100% (83-86 min) gradient of mobile phase B (95% ACN/0.125% FA) at 550 nL/min flow rate. Multi-notch MS$^3$-based TMT method coupled with Real-Time Search algorithm[70] was used for the analysis. The scan sequence started with MS$^1$ spectra analyzed by Orbitrap (resolution 120,000 at 200 Th, 400–1500 $m/z$, automatic gain control (AGC) target $2 \times 10^5$, maximum injection time 50 ms). Monoisotopic peaks were assigned, precursor fit filter was used (70% for a fit window of 0.5 Th), and dynamic exclusion window was applied (90 s, ±10 ppm). MS$^2$

spectra were analyzed by quadrupole-ion trap with collision-induced dissociation (Rapid scan rate, AGC $1.0 \times 10^4$, isolation window 0.5 Th, normalized collision energy (NBE) 34, maximum injection time 80 ms). Synchronous precursor selection (SPS) API-MS$^3$ scan collected top 10 most intense b- or y-ions matched with the real-time search algorithm[70]. MS$^3$ precursors were fragmented by high energy collision-induced dissociation (HCD) and analyzed with the Orbitrap (NCE 45, AGC $2.5 \times 10^5$, maximum injection time 200 ms, resolution 50,000 at 200 Th). Closeout was set at two peptides per protein for each fraction.

RAW files were converted to mzXML files. The searching database was constructed from Swiss-Prot human database (released on Jun 17, 2020 at UniProt), which was appended with common contaminants and reversed for target-decoy false discovery rate (FDR) estimation. Database searching was done with a 20-ppm precursor ion tolerance and 1.0005 Da product ion tolerance. Static modifications included carbamidomethylation at cysteine (+57.021 Da), TMT labeling at lysine (+229.162 Da for TMT or +304.207 Da for TMTpro) while variable modifications included oxidation at methionine (+15.995 Da) and TMT labeling at peptide N-termini (+229.162 Da for TMT or +304.207 Da for TMTpro). Peptide-spectrum matches (PSM) were filtered using a linear discriminant analysis algorithm while considering XCorr, ΔCn, missed cleavages, peptide length, charge state, and precursor mass accuracy. Identified peptides were controlled at 1% false discovery rate (FDR). Protein assembly was done by parsimony principle, and 1% FDR was applied to the protein level. For reporter ion quantification, signal-to-noise (S/N) ratio for each TMT channel was extracted with an integration tolerance of 0.003 Da. Proteins were quantified by summing the reporter ion counts across all matching PSMs. S/N of each channel was adjusted using the isotopic impurity table of TMT reagents provided by the vendor.

Protein quantitation values were exported and analyzed by R 3.6.3. Protein abundances were normalized according to the total reporter values in each channel assuming equal amount of loading. Pearson's correlation coefficient was calculated by the basic R function, *cor()*. For the classification by subcellular locations, proteins annotated as "very high" or "high" from Itzhak et al.[32] was used.

**Endosome and lysosome organellar proteomics.** Proteomics of purified lysosomes and endosomes was performed as described (https://doi.org/10.17504/protocols.io.bys6pwhe). Lysosomal and endosomal fractions purified as described above were first reduced by addition of TCEP to 5 mM final and incubated at 25 °C for 30 min. Cysteines were alkylated by addition of iodoacetamide to 15 mM final and incubated at 25 °C for 30 min and protected from light. Samples were diluted with EPPS buffer for 1 M urea final concentration. Proteins were precipitated by addition of 6.1 N TCA solution to 20% final and incubation at 4 °C for 1 h. Samples were centrifuged at $20,000 \times g$ for 15 min at 4 °C, and supernatants were removed. Samples were washed twice with ice-cold acetone by centrifuging at $20,000 \times g$ for 10 min at 4 °C. After final wash, the protein pellets were briefly dried in a SpeedVac. Pellets were resuspended in 5 µL of 8 M urea buffer followed by sonication in a water bath sonicator, and urea was diluted by addition of 5 µL of 200 mM EPPS. Peptides were digested by addition of 0.2 µg of LysC and incubated at 30 °C for 4 h. Urea was further diluted to 1.6 M final by addition of 200 mM EPPS. Peptides were further digested by addition of 0.2 µg trypsin and incubated at 37 °C overnight. The next day, ACN was added to 30% final, and the peptides were reacted with 10 µL of the TMTpro reagents (12.5 µg/µL in anhydrous ACN) for 1 h at 25 °C. Labeling was quenched by addition of hydroxylamine to 0.5% final followed by incubation at room temperature for 15 min. Pooled sample was dried and pre-fractionated using Pierce High-pH Reversed-Phase Peptide Fractionation Kit following the manufacturer's instructions. Gradient eluates were concatenated to final 4 fractions followed by desalting with C18 StageTip. In some experiments, cells were serum starved for 1 h, prior to addition

of TF (25 µg/mL), and cells harvested for Endo-IP at the indicated times followed by immunoblotting or proteomics.

Mass spectrometry was performed by an Orbitrap Eclipse coupled with a Proxeon EASY-nLC1200 liquid chromatography pump. Peptides were separated on a microcapillary column (100 µm inner diameter) packed with ~35 cm of Accucore150 resin (2.6 µm, 150 Å, Thermo Fisher Scientific) with 5–30% (3–108 min), 30–99% (108–113 min) gradient of mobile phase B (95% ACN, 0.125% of FA) at 550 nL/min flow rate. FAIMS Pro Interface and multi-notch MS$^3$-based TMT method coupled with Real-Time Search algorithm was used for the analysis. The scan sequence started with MS$^1$ spectra analyzed by Orbitrap (resolution 120,000 at 200 Th, 400–1500 m/z, automatic gain control (AGC) target $4 \times 10^5$, maximum injection time 50 ms). Monoisotopic peaks were assigned, precursor fit filter was used (70% for a fit window of 0.5 Th), and dynamic exclusion window was applied (120 s, ±7 ppm). Precursor ions were selected using a cycle type of 1.25 s/CV with FAIMS CV of −40/−60/−80. MS$^2$ spectra were analyzed by quadrupole-ion trap with collision-induced dissociation (Rapid scan rate, AGC $1.0 \times 10^4$, isolation window 0.5 Th, normalized collision energy (NBE) 34, maximum injection time 86 ms). Synchronous precursor selection (SPS) API-MS$^3$ scan collected top 10 most intense b- or y-ions matched with the real-time search algorithm[70]. MS$^3$ precursors were fragmented by high energy collision-induced dissociation (HCD) and analyzed with the Orbitrap (NCE 45, AGC $2.5 \times 10^5$, maximum injection time 200 ms, resolution 50,000 at 200 Th). Closeout was set at two peptides per protein for each fraction.

RAW files were converted to mzXML files. The searching database was constructed from Swiss-Prot human database (released on Jun 17, 2020 at UniProt), which was appended with common contaminants and reversed for target-decoy false discovery rate (FDR) estimation. Searches were done with a 20-ppm precursor ion tolerance and 1.0005 Da product ion tolerance. Static modifications included carbamidomethylation at cysteine (+57.021 Da), TMT labeling at lysine (+229.162 Da for TMT or +304.207 Da for TMTpro) while variable modifications included oxidation at methionine (+15.995 Da) and TMT labeling at peptide N-termini (+229.162 Da for TMT or +304.207 Da for TMTpro). Peptide-spectrum matches (PSM) were filtered using a linear discriminant analysis algorithm while considering XCorr, ΔCn, missed cleavages, peptide length, charge state, and precursor mass accuracy, as done previously[30]. Identified peptides were controlled at 1% false discovery rate (FDR). Protein assembly was done by parsimony principle, and 1% FDR was applied to the protein level. For reporter ion quantification, signal-to-noise (S/N) ratio for each TMT channel was extracted with an integration tolerance of 0.003 Da. Proteins were quantified by summing the reporter ion counts across all matching PSMs.

Protein quantitation values were exported and analyzed by R 3.6.3. Two-sided Student's t-test was performed by *t_test()* function in the *rstatix* package version 0.7.0. Individual p values were adjusted for multiple testing correction, which was done by *mt.rawp2adjp()* function with two-stage Benjamini & Hochberg (2006) step-up FDR-controlling procedure in the *multtest* package version 2.42.0.

**APP/Aβ TOMAHAQ proteomics.** TOMAHAQ experiments were performed using the Tomahto software package[30] on a Thermo Scientific Orbitrap Eclipse Tribrid mass spectrometer coupled to an Easy-nLC 1200 UHPLC system as described in dx.https://doi.org/10.17504/protocols.io.bys8pwhw. Each sample was separated on an in-house packed C18 column (30 cm, 2.6 µm Accucore [Thermo Fisher], 100 µm I.D.), and eluted using a 150-min method over a gradient from 5% to 38% B (95% ACN/0.125% FA). The instrument method only controlled Orbitrap MS$^1$ scans (resolution at 120,000; mass range 300 – 1500 m/z; automatic gain control (AGC) target $2 \times 10^5$, maximum injection time 50 ms). Peptide targets were imported into Tomahto, and the following decisions were made by Tomahto in real-time:

1. Tomahto listened to each collected MS$^1$ scan.
2. When a precursor ion matched *m/z* of a potential trigger peptide (Supplementary Data 7) (±10 ppm mass accuracy; matched charge state; minimal intensity of $5 \times 10^4$), Tomahto prompted insertion of an Orbitrap MS$^2$ scan (Trigger MS$^2$) with the trigger peptide's precursor *m/z* (0.5 m/z isolation window; resolution at 15,000; AGC target $1 \times 10^4$; max injection time 120 ms; CID collision energy 35). Once collected, a real-time peak matching strategy (RTPM) was used to confirm the identity of the trigger peptide (must match > 6 fragment peaks within ±10 ppm).
3. If the trigger MS$^2$ was successfully matched, Tomahto prompted the insertion of an Orbitrap MS$^2$ scan (Target MS$^2$) using the target peptide m/z (Supplementary Data 7) (0.5 m/z isolation window; resolution at 15,000; AGC target $1 \times 10^5$; max injection time 900 ms; CID collision energy 35.1). The target peak m/z is a mixture of multiplexed endogenous peptides. At the same time, the MS$^2$ fragment ions and their intensities for the trigger MS$^2$ were stored in memory as a template library spectrum. After collection, the target MS$^2$ scan was used to confirm that the target peptide was present at levels sufficient for detection. This was accomplished via RTPM where fragment ions must be present in the spectrum (±10 ppm) and rank ordered by intensity from the trigger MS$^2$. SPS fragment ions were now selected from this scan. Only *b*- and *y*-type ions were considered for selection provided they had TMT modifications. SPS candidates were required to match the fragmentation pattern of the stored library spectrum, meaning fragment ratios relative to the highest fragment were within ± 50% of that in the stored spectrum. In addition, each SPS candidate underwent a purity filter of 0.5 (at least 50% of the signal attributed to the fragment ion within a 3 *m/z* window) to be included in the final list.
4. Upon confirmation of target peptide presence and successful selection of SPS ions, Tomahto next triggered an ion trap SPS-MS$^3$ prescan (normal scan mode; AGC target of $1 \times 10^6$; max injection time of 10 ms). This was used to quickly estimate the signal strength for the TMT reporter ions. This estimate was used to set the lengthy injection times needed for the SPS-MS$^3$ scan detection in the Orbitrap.
5. Following the prescan, Tomahto prompted the insertion of the SPS-MS$^3$ quantification scan (resolution of 50,000; SPS ions from part 2, 0.5 *m/z* window, max injection time of 5000 ms).

Raw data were processed by the data analysis module of Tomahto. RawFileReader (https://planetorbitrap.com/rawfilereader) was used to read files, and spectra were matched to synthetic trigger peptide or endogenous target peptides, respectively.

Summed signal-to-noise ratio (S/N) were exported to csv file and analyzed by R 3.6.3. S/N of each channel was adjusted using the isotopic impurity table of TMT reagents provided by the vendor. Adjusted S/N values were then normalized according to the total reporter values in each channel according to a SPS-MS$^3$ analysis, assuming equal amount of loading. Statistical significance between DMSO- and compound-treated group was tested using two-sided Student's *t*-test with *t_test()* function in the *rstatix* package version 0.7.0.

For the absolute quantification of the target peptides, selected ion monitoring (SIM) experiments were used. SIM experiments were performed on a Thermo Scientific Orbitrap Eclipse Tribrid mass spectrometer coupled to an Easy-nLC 1200 UHPLC system. Each sample was separated on an in-house packed C18 column (30 cm, 2.6 um Accucore [Thermo Fisher], 100 μm I.D.), and eluted using a 150-min method over a gradient from 5% to 38% B (95% ACN, 0.125% FA). Target peptides were monitored within a 50 min window around the scheduled retention time (Supplementary Data 7). A pair of trigger and target peptides were isolated and accumulated separately targeting same AGC value ($5 \times 10^4$; resolution at 240,000) and subsequently analyzed

in a single Orbitrap SIM scan. If two target peptides share similar retention time and same FAIMS CV values, their detections were further multiplexed into a single SIM scan, as indicated by SIM ID in Supplementary Data 7.

RAW files from the SIM experiments were imported into Skyline v20.2, and the precursor ion peaks were extracted with 10 ppm accuracy. Because the TMT-labeled target peptides and TMTsh-labeled trigger peptides have the same retention time, the area under each peak was measured, and the ratio between target and trigger was calculated. Absolute amount the target peptide was derived by multiplying the ratio by the known amount of the trigger peptide. The absolute amount of target peptide was then divided by the relative quantitation from TOMAHAQ to calculate the absolute amount from each channel.

### Statistics and reproducibility
Data analysis was performed as described in the appropriate experimental section above. Unless stated otherwise all quantitative experiments were performed in triplicate and average with standard error of the mean (SEM) reported. Representative data are shown in Figs. 1b, d, e, 3b, 5c, e, f, 6c, 7d; Supplementary Fig. 1a, b, d, g, 2b, 3a–c, 5a–d, g. Same results were observed at least twice for Western blots and image analysis.

### Reporting summary
Further information on research design is available in the Nature Research Reporting Summary linked to this article.

## Data availability
All MS raw files have been deposited in MassIVE with the identifiers MSV000088132 (proteomics) [https://doi.org/10.25345/C5RN99] and MSV000088048 (lipidomics) [https://doi.org/10.25345/C5MC31]. Annotated lipid spectra are in Supplementary Data 15. Uncropped images can be found in Source Data 1. Source data for individual plots can be found in Source Data 2. The searching database for proteomics was constructed from Swiss-Prot human database (released on Jun 17, 2020 at UniProt, [https://ftp.uniprot.org/pub/databases/uniprot/previous_major_releases/release-2020_06/]) Source data are provided with this paper.

## Code availability
C# script to annotate LipiDex spectral matches can be found at https://github.com/coongroup/LipiDexSpectrumAnnotator and at https://doi.org/10.5281/zenodo.7086361.

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

## Acknowledgements
This work was supported by Aligning Science Across Parkinson's (ASAP) initiative (to J.W.H.), NIH (R01NS083524, R01NS110395 to J.W.H., K01DK098285 to J.A.P., and P41GM108538 to J.J.C.), and a generous gift from Ned Goodnow (J.W.H.). The Michael J. Fox Foundation administers the grant ASAP-000282 on behalf of ASAP and itself. For the purpose of open access, the author has applied a CC-BY public copyright license to the Author Accepted Manuscript (AAM) version arising from this submission. H.P. was the recipient of a postdoctoral fellowship from the Edward R. and Anne G. Lefler Center for the Study of Neurodegenerative Disorders. We also thank the Nikon Imaging Center at Harvard Medical School and Harvard Medical School Electron Microscopy Facility for microscopy support, and Tomas Kirchhausen for helpful discussions.

## Author contributions
H.P. and J.W.H. conceived the study. H.P. performed proteomics, cell line construction, biochemical assays, cell biology, and informatic analyses. F.V.H. performed biochemistry, proteomics, and cell biology. J.C.P. performed biochemistry and cell biology. Q.Y. provided informatics support for Tomahto and experimental design. J.A.P. provided proteomics support under supervision of S.P.G. Lipidomics was performed by K.A.O., D.R.B., and L.S. under supervision of J.J.C. S.S. performed gene editing. The manuscript was written by J.W.H., H.P., and F.V.H. with input from all authors.

## Competing interests
J.W.H. is a consultant and founder of Caraway Therapeutics and is a founding board member of Interline Therapeutics. J.J.C. and S.P.G. are consultants for Thermo Fischer Scientific. The remaining authors declare no competing interests.
