## [Peer Review File · Nature Communications]

Spatial Snapshots of Amyloid Precursor Protein
Intramembrane Processing via Early Endosome ProteomicsEditorial Note: This manuscript has been previously reviewed at another journal that is not operating a transparent peer review scheme. This document only contains reviewer comments and rebuttal letters for versions considered at *Nature Communications*.

REVIEWER COMMENTS

Reviewer #1 (Remarks to the Author):

The authors have addressed some concerns and added new experiments. New requested tabs have been added to the supplementary tables and a new figure of a Venn diagram in an extended data Figure. The requested Hydroxy Dynasore experiment to distinguish lipids internalized by endocytosis from resident lipids of endosome membranes has not been done. The authors are urged to do this experiment. It is not an issue of testing each internalized cargo for piggy-backing lipids, it is to get at the lipid makeup of the endosomal membrane. The authors have concluded that the mutant APP expression does not lead to ATF4 expression and further justifies the mutant in an extended data figure of an anti-CTF western blot.

The authors have added new experiments in Figures and extended data figures on transferrin uptake that are indicated in the manuscript.

Reviewer #3 (Remarks to the Author):

This is a manuscript previously reviewed by and transferred from another Nature journal. In essence, the authors have used an immunoprecipitation approach to isolate EEA1- early/sorting endosomes, and applied proteomics and lipidomics analytical approaches to confirm the authenticity of the membranes and further describe their biochemical composition (proteins and lipids). Proof-of-principle for the specificity of the isolation and usefulness of the approach was provided by the presence of the amyloid precursor protein (APP) and its processing by β - and γ -Secretases into amyloidogenic A β peptides.

The study is technically well done and the authors have addressed the main points raised by the referees, including new experiments such as the proteomics upon transferrin uptake, which has significantly improved the quality of the study. Nevertheless, the article would greatly benefit from a better explanation of the novelty and power of the techniques developed, as well as by better contextualizing and discussing the results. This is necessary because the technique per se is not novel and widely used in the field in the past 20-30 years. The choice of APP is also debatable as its processing in the endosomes has been described before. One wonders why out of the proteins detected the authors chose to make their case on a protein extensively studied, also in the context of the endosomes. For example, the authors explain the results from the lipidomics in the context of previous work from the Gruenberg lab, when analyzing the abundance of BMPs in different populations of endosomal membranes. However, the new results are incremental and do not provide new insights that can be leveraged experimentally. The study provides a technology basis with the potential to identify novel mechanisms, but this potential is not yet harnessed.

In summary, this is a well done and careful study but that lacks technical novelty and novel inspiring findings.

I would also recommend addressing the following issues:

1. In the rebuttal letter, the authors pointed at the challenge of achieving a balance between efficiency and purity of organelle purification. The manuscript successfully addresses the purity, however, In order to know to which extent this balance is achieved with Endo-IP, the authors should also report the efficiency of EEA1 pulldown, and how it compares to the one of TMEM192 in Lyso-IP.

2. Regarding the validity of the lipidomics quantification without internal standards used in this manuscript, I would agree with the reservations by Referee.

3. Schematics in Figure 4 are far too detailed. I understand that in Fig. 4a, the authors aim to summarize the amyloidogenic and non-amyloidogenic pathways of APP, but considering that this is not a review article, the schematics should contain only essential information needed to understand the results of the experiments. Relevant information would be the organelles where the different Secretases localize, where they may cleave APP and which peptides would be produced by this cleavage. Details such as the localization of AP-4, Retromer, ESCRT-III or the Glycosylation sites of APP are just confounding factors here.

Similarly, in Fig. 4e, the sequences of each trigger peptide could be removed from the main figure, and reported in a separated table.

4. As requested, the authors have explained the inconsistency when measuring A β 34 in lines 465-467: "Routinely, signal-to-noise values for the A β 34 peptide did not pass a p-value cut-off for significantly changing with BSI or GSI treatment when compared with cells lacking APP (Fig. 6f), and we are therefore routinely unable to quantify this peptide." Despite this, they concluded that they are routinely unable to quantify this peptide, but later they state that A β 34 abundance is largely unchanged. Lines 497-498: "In contrast, the abundance of A β 34 was largely unchanged by this GSM (Fig. 7c,d) "

As the methodology does not allow to reliably quantify A β 34 peptide, conclusions cannot be drawn about this peptide. This is a weak point of this study.

5. The authors do not provide data of APP processing in neurons, which would be a more disease relevant cellular system. They may want to clearly state the limitations of the interpretation of their results in a pathophysiological context in the discussion.

Finally, this manuscript is more suitable for publication in a specialized membrane traffic/proteomics journal.

Reviewer #5 (Remarks to the Author):

The revised version provides additional details required for evaluation of the quality of the lipidomic analysis. However, there are still substantial issues:

- The authors responded that "product ions used for the identification are already included for each raw file in the repository on MassIVE; these are provided as .csv files". Because the ions used for identification are essential to evaluate the appropriate annotation of the lipid molecules, these data should be made easily accessible in the suppl. tables.
- The nomenclature was amended – Cer 18:1_18:0;O2 is incorrect, instead Cer 18:1;O2/18:0 would be correct if product ions justify N-acyl and/or sphingoid base.
- Free cholesterol as essential component of a mammalian lipidome is still missing. The authors replied: "Free cholesterol is not easily measured with electrospray ionization techniques used for the lipidomics analysis; see comments from Gallego et al.". Yes, it is true that it ionizes not very well – but the cited method shows a method for direct quantification. Additionally, there are also derivatization methods available for its quantification.
- The authors responded to the remark that biophysical properties of membranes are related to lipid composition: "Significantly changed peak areas for bulk classes of lipids would not likely occur by

chance, and this suggests the fold-change differences we have measured are indeed due to true differences in lipid composition between control and endo-IP material.” I do not believe that there may be relevant changes in lipid species – however biological interpretation needs mol and thus composition of lipids for biological interpretation (see also comment below).

- Extended Data Fig. 3e. shows several species which are quite uncommon for cellular lipidomes e.g. very long chain species PC 42:0, 44:0, 48:7. Such findings need to be proven by respective diagnostic production ions. Moreover, there are three PC O-35:0, which need to be reported as molecular species if their identification could be justified?

A quick check of the supplementary data revealed several quality issues, as for example:

- Analysis is not systematic – some species of a lipid class are reported in positive ion mode, some in negative ion mode; e.g. three LPC 18:1 were reported at 1.807, 1.882, 1.972 min retention time (RT) two in positive ion mode, one in negative ion mode; PC 34:0 is detected at 10.101 min (negative) and 10.814 min (positive). I would expect that both positive and negative ions are detected for the same species which could be used to justify their identification and regulation. Especially all PC and LPC species should be observed in positive ion mode.

- PC 41:1 was reported at 13.227 min (negative, m/z 856.68884) and 13.426 min (positive, m/z 858.70166). While the ion reported in positive ion mode fits to $[M+H]^+$, the negative ion mode ion $[M-H]^-$, does not exist due to the quaternary ammonium function of PC.

- Data do not fit to effective carbon number retention time (ECN) model (see attached tables and figures): Four PC 34:1 (RT 9.884 to 11.584), four PC 35:1 (RT 9.378 to 12.131) and four PC O-35:0 (RT 11.048 to 12.226) show a huge spread in RT, which are hardly explainable by structural variations.

- PG 18:1_18:1 was reported at RT 7.603 and 8.384 min and PG 16:0_16:0 at RT 7.483 and 8.261 min the later is impossible!

- A mass tolerance of 10 ppm was allowed for the precursor ions. The quick check revealed the PC 37:1 at 10.33 min m/z 802.64142 shows 11.7 ppm mass deviation from the target m/z 802.632032. In general, for lipidomics data such mass tolerance is too high considering that Type-II overlap in double bond series show only a m/z difference of 9 mDa.

In summary, the lipidomic analysis of the present study is not state of the art. Only quantitative data permit calculation of lipid composition, which is related to membrane biophysics and biological function, which should be the goal in such high-ranking publications. There are methods for quantification which cover the main species of cellular lipidomes e.g. by either HILIC or direct infusion methods. For these methods internal standards are available. Instead, the authors performed untargeted analysis by RPLC that has of course advantages concerning identification but are obviously prone to over-reporting as exemplified above.

ID	Class	Lipid Maps Abbrv.	Retention Ti	ECN	Quant Ion	Polarity
PC 28:1	PC	PC 28:1	7.445	28	676.49121	+
PC 29:1	PC	PC 29:1	7.842	29	690.50647	+
PC 14:0_16:1	PC	PC 14:0_16:1	8.16	30	704.52179	+
PC 31:1	PC	PC 31:1	8.564	31	718.53748	+
PC 32:1	PC	PC 32:1	8.986	32	732.55304	+
PC 33:1	PC	PC 33:1	9.424	33	746.56897	+
PC 34:1	PC	PC 34:1	9.884	34	760.58478	+
PC 34:1	PC	PC 34:1	10.589	34	760.58771	+
PC 34:1	PC	PC 34:1	11.464	34	760.58002	+
PC 34:1	PC	PC 34:1	11.584	34	760.591	+
PC 35:1	PC	PC 35:1	9.378	35	774.60864	+
PC 17:0_18:1	PC	PC 17:0_18:1	10.166	35	774.60059	+
PC 35:1	PC	PC 35:1	10.371	35	774.60059	+
PC 35:1	PC	PC 35:1	12.131	35	774.60095	+
PC 36:1	PC	PC 36:1	10.867	36	788.61591	+
PC 36:1	PC	PC 36:1	11.592	36	788.62366	+
PC 37:1	PC	PC 37:1	10.33	37	802.64142	+
PC 18:1_19:0	PC	PC 18:1_19:0	11.137	37	802.63208	+
PC 37:1	PC	PC 37:1	11.373	37	802.6322	+
PC 37:1	PC	PC 37:1	13.179	37	802.63226	+
PC 38:1	PC	PC 38:1	11.882	38	816.64783	+
PC 39:1	PC	PC 39:1	14.156	39	830.66376	+
PC 18:1_22:0	PC	PC 18:1_22:0	12.924	40	844.67926	+
PC 41:1	PC	PC 41:1	13.227	41	856.68884	-
PC 41:1	PC	PC 41:1	13.426	41	858.70166	+
PC 42:1	PC	PC 42:1	13.941	42	872.71039	+
PC 18:1_25:0	PC	PC 18:1_25:0	14.434	43	886.72699	+
PC 44:1	PC	PC 44:1	14.906	44	900.74109	+
PC 45:1	PC	PC 45:1	15.387	45	914.75861	+

ID	Class	Lipid Maps Abbrv.	Retention Ti	C-Number	DB	Quant Ion	Polarity
Plasmanyl-PC O-28:0	Plasmanyl-PC	PC O-28:0	8.533	28	0	664.52948	+
Plasmanyl-PC O-29:0	Plasmanyl-PC	PC O-29:0	8.833	29	0	678.54358	+
Plasmanyl-PC O-16:0_14:0	Plasmanyl-PC	PC O-16:0_14:0	9.449	30	0	692.55878	+
Plasmanyl-PC O-16:0_15:0	Plasmanyl-PC	PC O-16:0_15:0	9.749	31	0	706.57458	+
Plasmanyl-PC O-31:0	Plasmanyl-PC	PC O-31:0	9.945	31	0	706.57391	+
Plasmanyl-PC O-32:0	Plasmanyl-PC	PC O-32:0	10.434	32	0	720.59009	+
Plasmanyl-PC O-33:0	Plasmanyl-PC	PC O-33:0	10.735	33	0	734.60608	+
Plasmanyl-PC O-33:0	Plasmanyl-PC	PC O-33:0	11.19	33	0	734.6145	+
Plasmanyl-PC O-34:0	Plasmanyl-PC	PC O-34:0	10.954	34	0	792.61292	-
Plasmanyl-PC O-34:0	Plasmanyl-PC	PC O-34:0	11.477	34	0	748.62134	+
Plasmanyl-PC O-35:0	Plasmanyl-PC	PC O-35:0	11.048	35	0	806.62762	-
Plasmanyl-PC O-35:0	Plasmanyl-PC	PC O-35:0	11.77	35	0	762.63794	+
Plasmanyl-PC O-35:0	Plasmanyl-PC	PC O-35:0	12.009	35	0	762.63873	+
Plasmanyl-PC O-35:0	Plasmanyl-PC	PC O-35:0	12.226	35	0	762.64618	+
Plasmanyl-PC O-36:0	Plasmanyl-PC	PC O-36:0	12.533	36	0	776.65295	+
Plasmanyl-PC O-37:0	Plasmanyl-PC	PC O-37:0	12.833	37	0	790.66913	+
Plasmanyl-PC O-37:0	Plasmanyl-PC	PC O-37:0	13.054	37	0	790.66913	+
Plasmanyl-PC O-37:0	Plasmanyl-PC	PC O-37:0	13.253	37	0	790.67743	+
Plasmanyl-PC O-38:0	Plasmanyl-PC	PC O-38:0	13.573	38	0	804.68494	+
Plasmanyl-PC O-39:0	Plasmanyl-PC	PC O-39:0	14.109	39	0	818.70099	+
Plasmanyl-PC O-39:0	Plasmanyl-PC	PC O-39:0	14.25	39	0	818.71027	+
Plasmanyl-PC O-40:0	Plasmanyl-PC	PC O-40:0	14.579	40	0	832.7157	+

Reviewer #1 (Remarks to the Author):

The authors have addressed some concerns and added new experiments. New requested tabs have been added to the supplementary tables and a new figure of a Venn diagram in an extended data Figure. The requested Hydroxy Dynasore experiment to distinguish lipids internalized by endocytosis from resident lipids of endosome membranes has not been done. The authors are urged to do this experiment. It is not an issue of testing each internalized cargo for piggy-backing lipids, it is to get at the lipid makeup of the endosomal membrane. The authors have concluded that the mutant APP expression does not lead to ATF4 expression and further justifies the mutant in an extended data figure of an anti-CTF western blot.

The authors have added new experiments in Figures and extended data figures on transferrin uptake that are indicated in the manuscript.

We thank the reviewer for noting the additions we made to the experiments. The review correctly noted that that we have not performed the Dynasore experiment suggested. Given likely rapid intermixing of endosomal membranes from both clathrin-dependent (i.e. blocked by Dynasore) and certain forms of clathrin-independent (i.e. immune to Dynasore), it is unclear whether the proposed experiment would actually discriminate between lipids internalized by endocytosis and “resident” lipids on endosomes. It seems like one would need a way to “freeze” multiple membrane trafficking systems that deliver membranes to the endosome in order to answer this question. We would therefore respectfully submit that this is beyond the scope of the paper.

Reviewer #3 (Remarks to the Author):

This is a manuscript previously reviewed by and transferred from another Nature journal. In essence, the authors have used an immunoprecipitation approach to isolate EEA1-early/sorting endosomes, and applied proteomics and lipidomics analytical approaches to confirm the authenticity of the membranes and further describe their biochemical composition (proteins and lipids). Proof-of principle for the specificity of the isolation and usefulness of the approach was provided by the presence of the amyloid precursor protein (APP) and its processing by β - and γ -Secretases into amyloidogenic A β peptides. The study is technically well done and the authors have addressed the main points raised by the referees, including new experiments such as the proteomics upon transferrin uptake, which has significantly improved the quality of the study.

We appreciate the reviewer’s positive comments concerning the technical aspects of the paper as well as noting that we have addressed the main points in the prior review.

Nevertheless, the article would greatly benefit from a better explanation of the novelty and power of the techniques developed, as well as by better contextualizing and discussing the results. This is necessary because the technique per se is not novel and widely used in the field in the past 20-30 years.

As noted in the previous response to reviewers, we have not been able to identify papers that use EEA1 immunoprecipitation for direct and rapid isolation of endosomes. We wish that the authors had provided references for this method being used widely in the field. We were able to find one paper (PMID: 29523688) which performed a two-step isolation, first performing gradient fractionation of endosomes and followed by a second step using a-EEA1. The method, however, seems to be quite limited in the proteins that can be

identified (23 total trafficking and cargo proteins), which is ~90% less than our method achieves. Given that our method avoids the time-consuming gradient isolation and appears to be much better in terms of recovery of diverse endosomal proteins, we would submit that the method would be useful to the community. We also note that this paper (published in 2018) does not reference any prior studies on the use of immunoprecipitation for endosome isolation and also calls their method “newly developed”. We now mention this work in the DISCUSSION. When possible in the text, we also mention that our method allows endosome purification “directly from cell extracts”.

The choice of APP is also debatable as its processing in the endosomes has been described before. One wonders why out of the proteins detected the authors chose to make their case on a protein extensively studied, also in the context of the endosomes. For example, the authors explain the results from the lipidomics in the context of previous work from the Gruenberg lab, when analyzing the abundance of BMPs in different populations of endosomal membranes. However, the new results are incremental and do not provide new insights that can be leveraged experimentally. The study provides a technology basis with the potential to identify novel mechanisms, but his potential is not yet harnessed.

In summary, this is a well done and careful study but that lacks technical novelty and novel inspiring findings.

I would also recommend addressing the following issues: 1. In the rebuttal letter, the authors pointed at the challenge of achieving a balance between efficiency and purity of organelle purification. The manuscript successfully addresses the purity, however, In order to know to which extent this balance is achieved with Endo-IP, the authors should also report the efficiency of EEA1 pulldown, and how it compares to the one of TMEM192 in Lyso-IP.

We thank the reviewer for point this out. Previous work in the Sabatini lab that developed the Lyso-IP reported an efficiency of 3% based on measurements of lysosomal enzyme activity relative to whole cell extracts (Science, 2017). The extent of enrichment very much depends on several factors, including the amount of antibody resin used as well as the antibody incubation time. In practice, one would want to balance incubation time with what processes one might want to measure in subsequent steps. In order to address the reviewer’s comments, we now report the extent of depletion of EEA1 and TMEM92 under our standard 30 min incubations. We state in the methods:

” Under standard conditions including a 30 min immunoprecipitation, we recover ~2.5% of the EEA1 present in total cell extracts, which is similar to the 3% recovery of lysosomes as reported previously.¹²”.

2. Regarding the validity of the lipidomics quantification without internal standards used in this manuscript, I would agree with the reservations by Referee.

We address reviewer 5’s comments below.

3. Schematics in Figure 4 are far too detailed. I understand that in Fig. 4a, the authors aim to summarize the amyloidogenic and non-amyloidogenic pathways of APP, but considering that this is not a review article, the schematics should contain only essential information needed to understand the results of the experiments. Relevant information would be the organelles where the different Secretases localize, where they may cleave APP and which peptides would be produced by this cleavage. Details such as the localization of AP-4, Retromer, ESCRT-III or the Glycosylation sites of APP are just confounding factors here.

We respectfully have a difference of opinion here in that one of the goals of the aim of the schematic figure is to bring home the point that the proteomics in figures 2 and 3 are identifying proteins along the continuum of sorting endosomes – ranging from ESCRT, to retromer, to various sorting proteins, which is why these are displayed in the figure along with APP related cleavage enzymes and products. We referred to this figure extensively in the DISCUSSION when discussing the proteins identified by Endo-IP. We realize that Fig 4A comes subsequent to the relevant proteomics figures, but feel that the figure overall generally fits better in the context of APP, and do not see an easy way to move it forward in the figure presentation. As such, we have now indicated “(see Fig 4A below)” in the text when initially describing the continuum of sorting and MVB machinery that we identified in endosomes, and also maintain the reference to the figure within the DISCUSSION. We hope that this addresses the reviewer’s main concern. In terms of AP4 and Golgi, based on previous studies, this may actually be the dominant pathway trafficking of APP to endosomes, accounting for 90% of flux in some cells. Although we could remove this, it is nevertheless accurate in terms of the major trafficking system for APP.

Similarly, in Fig. 4e, the sequences of each trigger peptide could be removed from the main figure, and reported in a separated table.

All the peptides are present in a supplemental table. However, we agree that we can simplify the figure by excluding the sequences of the extracellular and C-terminal domain peptides and just indicating by small bars the number and identity of peptides used. Therefore we have re-drawn the figure to only include the details of the Ab peptides. We hope this addresses the reviewer’s main concern.

4. As requested, the authors have explained the inconsistency when measuring A β 34 in lines 465-467: "Routinely, signal-to-noise values for the A β 34 peptide did not pass a p-value cut-off for significantly changing with BSI or GSI treatment when compared with cells lacking APP (Fig. 6f), and we are therefore routinely unable to quantify this peptide." Despite this, they concluded that they are routinely unable to quantify this peptide, but later they state that A β 34 abundance is largely unchanged. Lines 497-498: "In contrast, the abundance of A β 34 was largely unchanged by this GSM (Fig. 7c,d) " As the methodology does not allow to reliable quantify A β 34 peptide, conclusions cannot be drawn about this peptide. This is a weak point of this study.

We had inadvertently not edited the “in contrast...” sentence. We have now removed this sentence from the text.

5. The authors do not provide data of APP processing in neurons, which would be a more disease relevant cellular system. They may want to clearly state the limitations of the interpretation of their results in a pathophysiological context in the discussion.

APP as well as the processing machinery is broadly expressed in many tissues, although for BACE, there is enrichment in brain tissues and pancreas as well as liver. As such, we expect that processing of APP would not be limited to brain tissue. Nevertheless, we have now added a sentence to indicate that further studies in neurons are required to establish whether the same processes are occurring in that biological setting: “It will be particularly interesting to explore APP processing using these approaches in neuronal cells in the future.”

Finally, this manuscript is more suitable for publication in a specialized membrane traffic/proteomics journal.

Reviewer #5:

We thank Reviewer #5 for their critique and careful analysis. Based on this critique we have reprocessed our data, manually evaluated spectra, and present in this revision a more highly curated set of annotations. Most notably, the fundamental biochemical conclusions made in the original manuscript are still highly supported by these updated data.

Namely to resolve the reviewers concern for potential false positive identifications based on a pooled search strategy and a 0.01 m/z MS1 search tolerance being too large, we reprocessed the original data. Specifically, we searched only immunoprecipitated samples (not including whole cell pooled controls), we narrowed the MS1 search tolerance to 0.005 m/z MS1 search tolerance, and limited our search to only the 'LipidEx Acetate HCD' library and a custom BMP lipid library. This more stringent reprocessing retained 264 lipid annotations from the original search, ~ 30 annotations were reported with different level of annotation (molecular species vs. species level annotation), six additional lipids were annotated, and four lipid identifications were modified to

more accurately reflect the MS/MS data collected. In total, the reprocessing step resulted in ~300 lipid annotations.

Next, we manually inspected these data and prepared ECN plots as requested by the reviewer. Following these steps, we removed ~ 30 of these annotations. Our responses to all specific concerns raised by the reviewer are detailed below. All the methods and supplementary data have been updated to reflect this reprocessing. Finally, we have redone the fold-change analysis presented in the manuscript (Figure 3 and Extended Data Figure 3). The updated Figure 3 is shown in

Rebuttal Figure 1 and, despite this reprocessing, the biological conclusions we reported initially have not changed.

Reviewer #5 (Remarks to the Author):

The revised version provides additional details required for evaluation of the quality of the lipidomic analysis. However, there are still substantial issues:

- The authors responded that “product ions used for the identification are already included for each raw file in the repository on MassIVE; these are provided as .csv files”. Because the ions used for identification are essential to evaluate the appropriate annotation of the lipid molecules, these data should be made easily accessible in the suppl. tables.

To facilitate even easier access to our identified tandem mass spectra, we have created a PDF document. This document, which is now included as a supplemental file, expands on Supplementary Data Table 6, indicates the identification, retention time, scan number, precursor m/z value, dot product score, and contains an annotated MS/MS spectrum. Below we paste an example spectrum (Rebuttal Figure 2) from the new supplemental file for a PC identification. We further utilized these annotated spectra to manually validate our software-based identifications – giving us greater confidence in our reported dataset.

Rebuttal Figure 2. Exemplary annotated spectra from new PDF compendium of identified spectra.

- The nomenclature was amended – Cer 18:1_18:0;O2 is incorrect, instead Cer 18:1;O2/18:0 would be correct if product ions justify N-acyl and/or sphingoid base.

Done. The products do justify the suggested annotation and we have updated this identification as requested.

- Free cholesterol as essential component of a mammalian lipidome is still missing. The authors replied: “Free cholesterol is not easily measured with electrospray ionization techniques used for the lipidomics analysis; see comments from Gallego et al.”. Yes, it is true that it ionizes not very

well – but the cited method shows a method for direct quantification. Additionally, there are also derivatization methods available for its quantification.

Rebuttal Figure 3. Cholesteryl sulfate abundances in control and Endo-IP samples.

We agree that Cholesterol is an important component of the mammalian lipidome. However, the manuscript does not contain conclusions based on cholesterol measurements and does not claim to be a complete compendium of the endosomal lipidome. Here, we only compare lipids which are detectable via discovery LC-MS/MS between ENDO-IP samples and control-IP samples.

That said, to address the reviewer’s comment, we employed the lipidomics core facility at Beth Israel Deaconess Medical Center, as we do not routinely measure cholesterol. From control and Endo-IP samples in triplicate (Rebuttal Figure 3), we detected cholesteryl sulfate, with ~three-fold enrichment in the Endo-IP sample. However, we did not add this data to the paper as we believe it adds no significance to the analysis. If, however, the editor/reviewer believe these data would be useful to the paper we are happy to consider including them.

- The authors responded to the remark that biophysical properties of membranes are related to lipid composition: “Significantly changed peak areas for bulk classes of lipids would not likely occur by chance, and this suggests the fold-change differences we have measured are indeed due to true differences in lipid composition between control and endo-IP material.” I do not believe that there may be relevant changes in lipid species – however biological interpretation needs mol and thus composition of lipids for biological interpretation (see also comment below).

While absolute quantification is necessary for certain analyses, global studies like this manuscript are commonly conducted using relative quantification. Specifically, we make no new claims regarding the biophysical properties of endosomal membranes which would require this type of precise and comprehensive absolute quantitation. Further, there are numerous peer-reviewed examples of relative lipid quantification being used for biological discovery. Notably, a recent work by this team that appeared in *Nature* in May 2022, leverages the exact same methodology and analysis pipeline to conduct relative quantification of lipids from hundreds of cell lysates with varying genetic background. These lipid measurements were key in revealing the functions of numerous mitochondrial proteins and were validated with extensive biochemical follow-up. Several others have implemented similar discovery lipidomics methods for relative quantification, for example:

Fiehn et al. Sci Data. 2018 (PMC6244184)

Saghatelian and Kahn et al. Cell. 2014 (PMC4260972)

Ortlund et al. Nature Cell Biol. 2022 (PMC9203275)

In another recent study using the discovery lipidomics methodology leveraged here, we described the identification of lipid QTL using the same relative quantification. Hundreds of lipid QTL were identified and mapped to gene loci with high confidence with numerous validated.

To the reviewer's point that "biological interpretation needs mol". We respectively push back on this statement. One need not look further than the entire fields of transcriptomics and proteomics as counter examples to this ideology. Even beyond mass spectrometry Western blotting and traditional imaging, generally do not provide absolute quantification, yet biologist have leveraged the relative quantification of these tools for decades with great success.

- Extended Data Fig. 3e. shows several species which are quite uncommon for cellular lipidomes e.g. very long chain species PC 42:0, 44:0, 48:7. Such findings need to be proven by respective diagnostic production ions. Moreover, there are three PC O-35:0, which need to reported as molecular species if their identification could be justified?

The reviewer offers a good suggestion to review the identification of these long chain species. The reprocessing step (described above) eliminated the PC 44:0 and PC 48:7 annotation. The PC 42:0 lipid identification, however, was retained. Manual inspection of this MS/MS spectrum confirms the annotation as PC 16:0₂₆:0 given that we see m/z 255 and m/z 395 ions in the negative mode MS2 and m/z 184 in the positive mode MS2 (Rebuttal Figure 4). However, the negative mode MS2 has a low signal-to-noise ratio which lowered the dot product score used for automated spectral identification (dot product 378 and reverse dot product of 400). From this spectral evidence, we can be very confident in the PC 42:0 annotation but, out of an abundance of caution, have left the identification at the species level.

Rebuttal Figure 4. Chromatograms and MS/MS spectra for the lipid annotated as PC 42:0. These data confirm our identification and suggest the acyl chains are 16:0 and 26:0.

For the three PC O-35:0 annotations, we leverage evidence from the positive mode MS2 (m/z 184 indicates phosphocholine head group) and MS1 mass. When we narrowed our MS1 search tolerance to from 0.01 Da to 0.005 Da, we found that one of the features that was previously identified as PC O-35:0 is no longer annotated as such. The two remaining features annotated as PC O-35:0 are highly correlated ($R^2 = 0.97$); given this, we have removed the feature at 12.0 min, and retain only one PC O-35:0 annotation.

A quick check of the supplementary data revealed several quality issues, as for example:

- Analysis is not systematic – some species of a lipid classes are reported in positive ion mode, some in negative ion mode; e.g. three LPC 18:1 were reported at 1.807, 1.882, 1.972 min retention time (RT) two in positive ion mode, one in negative ion mode; PC 34:0 is detected at 10.101 min (negative) and 10.814 min (positive). I would expect that both positive and negative ions are detected for the same species which could be used to justify their identification and regulation. Especially all PC and LPC species should be observed in positive ion mode.

We will add clarity here and in the method section regarding how the software pipeline selects ions for quantitation. The ion in the supplementary table for a given lipid is the ion used for relative quantification. The same ion is used to consistently quantify the specific lipid in all samples where it is detected. The particular quantitation ion is selected as the ion which is most consistently observed across all samples. In this way, we perform quantitation in a manner which best reflects the actual data collected and not based on pre-defined rules of what adducts or polarities should be most intense, as these intensities are specific to a given LC-MS/MS method setup. In our experience, this approach reduces the number of quantitative values which are generated via the gap-filling algorithm in Compound Discoverer – giving us greater confidence in our relative quantitation. Using this approach, only one ion for a specific lipid feature would be represented in our relative quantification table (either positive or negative, but not both) as additional adducts, isotopes, or in source fragments are also detected and excluded from quantification, see also LipiDex manuscript PMID: 29705063. Although a single ion is used for relative quantification of a specific lipid, the lipid identifications leverage all suitable evidence about an MS1 feature – including MS/MS in positive and negative. To clarify the specific annotations, we have now added a Supplementary Data File (PDF) with all evidence for identification of features per response to point #1.

Concerning the use of negative mode for quantitation, the reviewer is correct that all LPCs and PCs are typically detected in positive mode, and we would expect that positive mode would result in consistent quantification. Notably the LPC 18:1 features were detected in both positive mode and negative mode. These LPC features elute very early in our chromatographic separation with many other lyso species. Given the limitations to the instrument's MS/MS sampling rate, we do not sample every feature in both positive and negative mode. In this case, one of the LPC 18:1 features was more consistently identified in negative mode and, as such, the negative mode ion was used for quantitation for that lipid across all samples. These early-eluting lyso-lipid features often display peak splitting due to the large polarity difference between the starting mobile phase composition and the sample resuspension solvent. Given that these duplicate features' peak areas were well correlated, we have only retained one annotated LPC 18:1 (the largest of the peaks was retained) in our supplemental table.

Regarding the PC 34:0, after reprocessing the data we have removed the annotation at 10.101 min (negative). PC 34:0 remains in our dataset and the relative quantification is based on the positive mode data. Finally, per the reviewer's recommendation we have thoroughly evaluated the data for negative mode/positive mode quantified peaks in the revised table.

- PC 41:1 was reported at 13.227 min (negative, m/z 856.68884) and 13.426 min (positive, m/z 858.70166). While the ion reported in positive ion mode fits to $[M+H]^+$, the negative ion mode ion $[M-H]^-$, does not exist due to the quaternary ammonium function of PC.

Thank you for evaluating these data. At 13.227 we have overlapping m/z features in negative mode at m/z 856.68884 and positive mode at m/z 858.6946 (Rebuttal Figure 5). Given the overlap, Compound Discoverer associated these features to the same molecular weight ion and

linked these features to the spectral matching positive mode PC 41:1 annotation (MS2 + of 585.696 at RT 13.294, scan# 2553 of ENDO_IP_1 file). However, we agree with the reviewer that the negative mode 856.6884 is not likely PC 41:1 due to the quaternary ammonium functionality. We have removed the annotation from this feature.

Rebuttal Figure 5. Chromatograms linked to the annotation of lipid PC 41:1. The negative mode chromatogram here is unexpected for a PC 41:1 and we agree that these features were incorrectly associated when they are in fact not.

- Data do not fit to effective carbon number retention time (ECN) model (see attached tables and figures): Four PC 34:1 (RT 9.884 to 11.584), four PC 35:1 (RT 9.378 to 12.131) and four PC O-35:0 (RT 11.048 to 12.226) show a huge spreads in RT, which are hardly explainable by structural variations.

A majority of our data fit an effective carbon number retention time (ECN) model well. Rebuttal Figure 6 presents these data for all lipid annotations following the more conservative searching parameters used in this revision. The right panel of this figure plots all annotations after a further round of manual inspection, including inspection of all specific concerns raised in the last review. Specifically, we have reevaluated our data for ECN conformity and have removed 19 identifications that did not fit the ECN model and features identifications that were likely driven by co-fragmentation. Notably one of these features was misidentified as a PC when manual interpretation of the MS/MS evidence suggests that it is in fact a PE (this was one of the examples noted by the reviewer). The ECN plots for our data are shown below either before or after the reprocessing and manual validation. Finally, we note that it is established in the literature that lipid identifications can deviate from the ECN model. Rebuttal Figure 7 documents that isomeric lipids having different double bond locations can have significantly different retention times in reversed-

phase LC (White et al, PMID 35157429). Thus, it is possible that some of the removed and filtered identifications based on this ECN approach were legitimate. That said, to satisfy the reviewer we have removed them.

Rebuttal Figure 6. Effective carbon number retention time (ECN) plots of lipidomic data before and after more conservative filtering and manual inspection. Left panel presents the ECN plot of the lipid identifications included in this revision after the more conservative searching parameters described above. The right panel presents the ECN plot of the lipid annotations after further manual inspection and filtering.

Figure S1: Elution of PC 18:1_18:1 in Reversed-Phase LC-MS. Elution of PC with the same acyl chain composition and differing double bond positioning in RP-LC-MS from 3 sequential injections of PC (18:1_18:1) standards with known stereochemistry. PC 18:1(9Z) elutes with complete baseline separation from 9E and 6Z counterparts. The 9E and 9Z species coelute

Rebuttal Figure 7. Figure from White et al. (PMID 35157429) that documents potential for retention time separation on a reversed-phase column based on double bond location. This manuscript documents that retention times can change for lipids having the same composition but varied location of double bonds. These data confirm that when using reversed-phase separations the ECN plots may vary from linearity without inherently containing incorrect annotations.

- PG 18:1_18:1 was reported at RT 7.603 and 8.384 min and PG 16:0_16:0 at RT 7.483 and 8.261 min the later is impossible!

We thank the reviewer for pointing out this concern. We have investigated these PG features, all PG features reported had dot product scores greater than 500 and were considered sufficient for annotation. However, after closer inspection, we noted that early eluting features annotated as PGs had lower dot product scores and signal-to-noise ratios than PGs at the later expected retention times. It is likely that the earlier eluting species identified as PGs resulted from in-source fragmentation of other lipid species. We have implemented a more strict score threshold for PGs, requiring greater than 700 dot product score and 700 reverse dot product score to be annotated as a PG. This stringent threshold reduced total number of PGs annotated from 16 to 8. Further, the remaining have been manually inspected and verified.

- A mass tolerance of 10 ppm was allowed for the precursor ions. The quick check revealed the PC 37:1 at 10.33 min m/z 802.64142 shows 11.7 ppm mass deviation from the target m/z 802.632032. In general, for lipidomics data such mass tolerance is too high considering that Type-II overlap in double bond series show only a m/z difference of 9 mDa.

For precursor ion matching Lipidex uses an m/z window cutoff. Our initial data had a 0.01 m/z window and in our reprocessed data presented here we reduced that to 0.005 m/z mass tolerance. This resulted in a mass error of less than 0.3 ppm on average. Further, we note that while HILIC and direct infusion methods do suffer from Type-II overlap due to co-elution (HILIC) or co-ionization (direct infusion) of lipid species from the same lipid class; RP-LC methods leverage chromatographic separations to reduce the spectra complexity and the likelihood of this isotopic overlap. Nonetheless, our new highly conservative mass tolerance of 0.005 m/z would eliminate any Type-II overlap that is present.

In summary, the lipidomic analysis of the present study is not state of the art. Only quantitative data permit calculation of lipid composition, which is related to membrane biophysics and biological function, which should be the goal in such high-ranking publications. There are methods for quantification which cover the main species of cellular lipids e.g. by either HILIC or direct infusion methods. For these methods internal standards are available. Instead, the authors performed untargeted analysis by RPLC that has of course advantages concerning identification but are obviously prone to over-reporting as exemplified above.

Although the reviewer's insightful analysis led us to re-analyze and greatly increase our confidence in the lipid identifications made, we kindly disagree with the reviewer that these data are not state of the art. RPLC is an alternative to direct infusion and HILIC methods which has shown to be ideally suited for discovery analysis and relative quantitation between samples for impactful studies (PMID: 30457571, PMID: 25303528, PMID: 35681008). All three major lipidomics methodologies (infusion, RPLC, and HILIC) have advantages and disadvantages for identification and quantitation – a major reason why all of these methods are commonly used by advanced lipidomics labs (PMID: 30830346). Direct infusion and HILIC also suffer methodological-specific problems, including overlap in in-source fragments with true features (with direct infusion), and increased spectra complexity which comes with increased concerns of ion suppression affecting quantitation and limits of detection. We have found that untargeted RPLC methods provide meaningful insight into biology (PMID: 34332123, PMID: 33096026, PMID: 35614220) and they offer an exploratory approach to find previously undiscovered molecules (PMID: 32958938).

Finally, regardless of whether the technology is state of the art or not, the data generated here has driven our understanding of endosomal biology and supported the basis of our endo-IP strategy.

REVIEWER COMMENTS

Reviewer #5 (Remarks to the Author):

The authors polished the lipidomic data and removed some of the exemplified false identifications, and now filtered more stringent. Moreover, some annotations were amended. However, the main issue of this reviewer is not addressed that only relative comparison but not quantitative data are provided.

The authors replied "To the reviewer's point that "biological interpretation needs mol". We respectively push back on this statement. One need not look further than the entire fields of transcriptomics and proteomics as counter examples to this ideology." Please do not cite my statement outside of its content. My statement was related to lipids but not to other biomolecules. Because biological roles/functions of other biomolecules are quite different from those of lipids, the authors conclusion makes no sense. Moreover, it is important to mention that lipidomics has provided quantitative data already for more than a decade. Thus, lipidomics should not be compared with other omics disciplines and the value of quantification called "ideology". Beside more insight, concentrations provide a huge advantage to make data comparable to other studies and to evaluate their reliability.

The authors argue that the methodology applied here could provide useful insights and cited several reports – yes, I agree that the applied approach could be useful depending on the research question. The strength of the applied method is to discover new lipids or to screen for unexpected roles of lipids in a specific context. In my opinion, both are not the case in the present study. Instead, this study would benefit from quantitative lipidomic data, which describe the detailed lipid composition of the different fractions. Finally, the lipidomic data represent only a small part of the message of the manuscript and even I could not find any statement on lipidomics results in the abstract. In general, the current study focuses on proteomics. Therefore, I recommend either to omit lipid data or to include quantitative data as a valuable resource.

From a quick check of the MS2 spectra, which were now included as data supplement, I have many questions concerning the quality of the data: I just would like to exemplify my concerns because I am not able to perform a full curation of the data set:

- DG species are identified as protonated ions e.g. Alkenyl-DG P-16:0_20:4 [M+H]⁺. Typically, this lipid class is detected in positive ion mode as ammonium adduct ion (also in a previous study of this group see Coon et al. Nature Metabolism. 2020, Extended Data Fig. 7). Did the authors check with a standard whether protonated ions are formed? The MS2 spectra show a NL of 20:4 which does not provide evidence for plasmalogen annotation – please see also comment below.

- The annotation of plasmalogens is questionable (see also the example above). For example the MS2 spectra for Plasmenyl-PC P-16:0_16:0 [M+Ac-H]⁻ and Plasmenyl-PE P-16:0_16:1 [M-H]⁻ only show acyl fragments justifying the acyl chain but not the plasmalogen bond which is a vinyl ether. Unless a proof of this bond type, these species should be annotated as alkyl species PC O-16:1_16:0 or PE O-16:1_16:1, respectively.

We respond to the specific questions of Reviewer #5 below.

REVIEWER COMMENTS

Reviewer #5 (Remarks to the Author):

The authors polished the lipidomic data and removed some of the exemplified false identifications, and now filtered more stringent. Moreover, some annotations were amended. However, the main issue of this reviewer is not addressed that only relative comparison but not quantitative data are provided.

The authors replied “To the reviewer’s point that “biological interpretation needs mol”. We respectively push back on this statement. One need not look further than the entire fields of transcriptomics and proteomics as counter examples to this ideology.” Please do not cite my statement outside of its content. My statement was related to lipids but not to other biomolecules. Because biological roles/functions of other biomolecules are quite different from those of lipids, the authors conclusion makes no sense. Moreover, it is important to mention that lipidomics has provided quantitative data already for more than a decade. Thus, lipidomics should not be compared with other omics disciplines and the value of quantification called “ideology”. Beside more insight, concentrations provide a huge advantage to make data comparable to other studies and to evaluate their reliability.

The authors argue that the methodology applied here could provide useful insights and cited several reports – yes, I agree that the applied approach could be useful depending on the research question. The strength of the applied method is to discover new lipids or to screen for unexpected roles of lipids in a specific context. In my opinion, both are not the case in the present study. Instead, this study would benefit from quantitative lipidomic data, which

describe the detailed lipid composition of the different fractions. Finally, the lipidomic data represent only a small part of the message of the manuscript and even I could not find any statement on lipidomics results in the abstract. In general, the current study focuses on proteomics. Therefore, I recommend either to omit lipid data or to include quantitative data as a valuable resource.

From a quick check of the MS2 spectra, which were now included as data supplement, I have many questions concerning the quality of the data: I just would like to exemplify my concerns because I am not able to perform a full curation of the data set:

- DG species are identified as protonated ions e.g. Alkenyl-DG P-16:0_20:4 [M+H]⁺. Typically, this lipid class is detected in positive ion mode as ammonium adduct ion (also in a previous study of this group see Coon et al. Nature Metabolism. 2020, Extended Data Fig. 7). Did the authors check with a standard whether protonated ions are formed? The MS2 spectra show a NL of 20:4 which does not provide evidence for plasmalogen annotation – please see also comment below.

The Alkenyl-DG example mentioned above, is one of six identifications in this class. We originally detected and annotated these compounds as protonated cations in experiments with HAP1 cells that were published along with our software tool for lipid annotations (see Hutchins et al. Cell Systems 2018, Hutchins et al. JASMS 2019). Those lipids were then included in our lipid libraries and resulted in the confident matches reported here. We have closely examined these species per the reviewer's suggestion. There appear to be no, or very few, co-eluting species that would confound identification. Accurate mass precursor searches of the Lipid Maps web tool produces no other viable candidates at the mass tolerances we used here. That said, the MS/MS spectra are of low S/N. Further, we have not yet identified a viable commercial standard in this class that could be used to confirm ionization preferences. **While we are quite confident that these are likely the correct annotations, we have chosen the most direct solution to remove the six identifications from our dataset thus there can be no ambiguity or concerns from the reviewer.** We have updated relevant figures and tables to reflect changes in the Alkenyl-DG identifications.

- The annotation of plasmalogens is questionable (see also the example above). For example the MS2 spectra for Plasmenyl-PC P-16:0_16:0 [M+Ac-H]⁻ and Plasmenyl-PE P-16:0_16:1 [M-H]⁻ only show acyl fragments justifying the acyl chain but not the plasmalogen bond which is a

vinyl ether. Unless a proof of this bond type, these species should be annotated as alkyl species PC O-16:1_16:0 or PE O-16:1_16:1, respectively.

Again, we are confident these annotations are correct; however, to the reviewer's concern we examined the 27 annotated plasmemyl species. Nine of them contain only acyl chain fragments, potentially adding ambiguity to the identification. Therefore, we have adjusted the annotation of these nine species from P- to O- as recommended by the reviewer.